# A curative combination cancer therapy achieves high fractional cell killing through low cross-resistance and drug additivity

Adam C Palmer[1†‡], Christopher Chidley[1†], Peter K Sorger[1,2]*

[1]Laboratory of Systems Pharmacology, Harvard Medical School, Boston, United States; [2]Department of Systems Biology, Harvard Medical School, Boston, United States

**Abstract** Curative cancer therapies are uncommon and nearly always involve multi-drug combinations developed by experimentation in humans; unfortunately, the mechanistic basis for the success of such combinations has rarely been investigated in detail, obscuring lessons learned. Here, we use isobologram analysis to score pharmacological interaction, and clone tracing and CRISPR screening to measure cross-resistance among the five drugs comprising R-CHOP, a combination therapy that frequently cures Diffuse Large B-Cell Lymphomas. We find that drugs in R-CHOP exhibit very low cross-resistance but not synergistic interaction: together they achieve a greater fractional kill according to the null hypothesis for both the Loewe dose-additivity model and the Bliss effect-independence model. These data provide direct evidence for the 50 year old hypothesis that a curative cancer therapy can be constructed on the basis of independently effective drugs having non-overlapping mechanisms of resistance, without synergistic interaction, which has immediate significance for the design of new drug combinations.

*For correspondence:
peter_sorger@hms.harvard.edu

[†]These authors contributed equally to this work

Present address: [‡]Department of Pharmacology, Computational Medicine Program, Lineberger Comprehensive Cancer Center, University of North Carolina at Chapel Hill, Chapel Hill, United States

## Introduction

The majority of cancers are treated with combination therapies. For some types of cancer, multidrug combinations have been developed that produce frequent cures, whereas cure by monotherapy is rare (*Frei and Antman, 2000*). In current practice, the search for new drug combinations focuses on identifying drugs that exhibit synergy. Although 'synergy' is often used loosely it is best defined by Bliss or Loewe criteria, which test whether a combination is stronger than expected from the sum of the drugs' individual effects; antagonism arises when combinations are less active than additivity would predict. In translational cancer biology such measurements are most commonly made using cultured cells or genetically defined mouse models. Despite the current emphasis on synergy, historically successful combinations were developed according to quite different hypotheses. For example, Law and Frei et al. argued for combining drugs that are independently effective and have non-overlapping mechanisms of resistance (*Frei et al., 1965*; *Law, 1956*; *Law, 1952*). Such combinations are expected to overcome clonal heterogeneity present within each patient's cancer. Heterogeneity between patients can also be a reason why drug combinations improve response rates, even when each patient only benefits from the most active monotherapy. However, cure is almost never achievable by monotherapy, and the superiority of curative combinations cannot be explained by patient variability in best single-drug response (the topic of our previous work [*Palmer and Sorger, 2017*]).

The pharmacological principles underlying curative combination therapies are largely unknown because most such combinations were developed via empirical experimentation in patients, and the combinations that worked have only rarely been subjected to detailed mechanistic analysis. Knowing

the design principles behind existing curative regimens should assist in rationally assembling new cancer medicines into curative combinations. A key question for such an effort, and for oncology drug discovery in general, is determining which among several different properties of a combination should be given the greatest weight: strong individual activity and low cross-resistance as proposed five decades ago, or synergistic interaction as currently emphasized (*Han et al., 2017*; *Nature Medicine, 2017*).

Testing whether a higher order multi-drug combination (that is, one with more than two constituents) exhibits synergistic interaction can only be accomplished ex vivo by measuring and analyzing the responses of cells to drugs applied individually and in combination over a range of concentrations, for example by isobologram analysis (*Greco et al., 1995*; *Loewe, 1953*). When evaluated at fixed doses the superiority in vivo of a combination over monotherapy can occur without a true pharmacological interaction and is therefore not sufficient evidence of synergy (*Berenbaum, 1989*). Testing whether a combination exhibits low cross-resistance is more challenging because it requires systematic exploration of resistance mechanisms; different mechanisms of resistance display different cross-resistance properties. It has long been possible to isolate cell clones resistant to single drugs and then assay for sensitivity to other drugs, but this approach is not practical at a scale needed to test Law and Frei's hypothesis, as was recognized by *Law (1956)*. Efficient analysis of cross-resistance has become feasible only recently with technical breakthroughs in multiplexed clone tracing and reverse genetic screening. DNA barcode libraries allow large numbers ($\geq 10^6$) of uniquely tagged clones to be tested in parallel for resistance to multiple drugs (*Bhang et al., 2015*), and genetic screens using CRISPR-Cas9 technologies enable genome-wide identification of loss and gain of function changes that confer resistance (*Bhang et al., 2015*; *Gilbert et al., 2014*; *Jost et al., 2017*; *Shalem et al., 2014*; *Wang et al., 2014*). To date, barcode and CRISPR-Cas9 libraries have been used to study mechanisms of resistance primarily to targeted therapies and identify new combinations of such drugs (*Bhang et al., 2015*; *Hata et al., 2016*); they have not yet been used to test Law and Frei's 'non-overlapping resistance' hypothesis by analyzing combinations of cytotoxic drugs that are the backbone of curative therapies.

In this paper, we measure pharmacological interaction and cross-resistance among components of R-CHOP, a five drug chemo-immunotherapy that achieves high cure rates in Diffuse Large B-Cell Lymphoma (DLBCL). R-CHOP has five constituents: R – rituximab, a humanized monoclonal antibody against CD20, a protein expressed on the surface of all B cells; C – cyclophosphamide (Cytoxan) an alkylating agent; H – hydroxydaunomycin (doxorubicin, or Adriamycin), a topoisomerase II inhibitor; O – Oncovin (vincristine), an anti-microtubule drug and; P – prednisone, a steroid. R-CHOP was developed over an extended period of time via clinical experiments in humans (*Lakhtakia and Burney, 2015*). The constituents of R-CHOP are known to be individually cytotoxic to DLBCL cells in vivo, and the drugs have largely non-overlapping dose-limiting toxicities, which permits their combined administration in patients. The reasons for the clinical superiority of R-CHOP in DLBCL remain poorly understood. *Pritchard et al. (2013)* observed no synergy among pairs of drugs in CVAD (similar to CHOP) in a mouse cell line model of Non-Hodgkin lymphoma, and in profiling the effects of 93 gene knockdowns by RNA interference on drug sensitivity, the change in sensitivity to CVAD was equal (for almost every knockdown) to the average of its changes in single-drug sensitivity; this demonstrates that CVAD does not act as a more potent version of a single drug, nor does it exhibit a new signature of genetic dependencies.

We tested for pharmacological interaction among all pairs of R-CHOP constituents across a full dose range in three DLBCL cell lines and assessed interaction using both the Bliss independence and Loewe additivity criteria. We observed little if any synergy: most drug pairs were additive and some were antagonistic. We also tested higher order combinations at fixed dose ratios with similar results. We then screened for cross resistant mutations using random mutagenesis with clone tracing as well as CRISPR interference (CRISPRi) and CRISPR activation (CRISPRa) with genome-scale libraries. The rate of multi-drug resistance was near the theoretical minimum predicted by *Law (1952)*, where the 'fractional killing' achieved by a combination is the product of each individual drug's fractional kill. This suggests that high single-agent activity and low-cross-resistance are key attributes of the curative R-CHOP regimen.

## Results

### Components of R-CHOP do not exhibit synergy in killing Diffuse Large B-Cell Lymphoma cells

Pharmacological interactions among R-CHOP constituents were measured in human Pfeiffer, SU-DHL-4 and SU-DHL-6 cell lines. All three lines are derived from germinal center B-like DLBCL, the subtype most responsive to R-CHOP (*Alizadeh et al., 2000*). Prednisone and cyclophosphamide are pro-drugs that are activated by liver metabolism. We therefore used the pre-activated forms of these drugs: prednisolone and 4-hydroperoxy-cyclophosphamide (which spontaneously converts to the active compound 4-hydroxy-cyclophosphamide in water) (*Ludeman, 1999*). Rituximab kills B-cell lymphomas through multiple CD20-dependent mechanisms that include complement-mediated cytotoxicity (CMC), antibody-dependent cell cytotoxicity (ADCC) and direct killing via CD20 cross-linking (*Weiner, 2010*). Consistent with previous reports (*Kobayashi et al., 2013*), we observed that rituximab can kill DLBCL cells in culture via CMC when human serum is included in the culture media (*Figure 1—figure supplement 1A*). Among seven DLBCL cell lines tested, none exhibited a cytotoxic response to prednisolone alone at clinically relevant concentrations, although the rate of cell division was reduced (*Figure 1—figure supplement 1B*). Prednisone is cytotoxic to DLBCL in first-line clinical care (*Lamar, 2016*); the absence of cytotoxicity in DLBCL cell culture, which is consistent with other studies (*Knutson et al., 2014*), might reflect selection for prednisone resistance in cell lines established from post-treatment patients. As there exist no generally available treatment-naïve DLBCL cell lines, we are not able to test whether such cultures might respond in vitro to prednisolone.

Pharmacodynamic interactions among drugs comprising R-CHOP were first measured in Pfeiffer cells. For each of 10 drug pairs, an 11 × 11 'checkerboard' was created with each drug increasing in concentration along one of the two axes, spanning a 100-fold range. Cells were incubated with drugs for 72 hr, which spans at least one in vivo half-life in humans for each of C, H, O and P (*de Jonge et al., 2005*; *Gidding et al., 1999*; *Speth et al., 1988*); R has an elimination half-life of 3 weeks in humans (*Tran et al., 2010*). Cell viability was measured using a luminescent ATP assay (Cell-Titer-Glo) that was linearly proportional to live cell number as determined by microscopy and vital staining (*Figure 1—figure supplement 1C*). The ratio of cell number in drug-treated and untreated control cultures (relative cell number) was used to compute normalized growth rate inhibition values (GR values [*Hafner et al., 2016*]) (*Figure 1—figure supplement 1D*). Pharmacological interaction was then assessed based on excess over Bliss Independence and by isobologram analysis (which tests for Loewe additivity [*Berenbaum, 1989*]). We have previously used isobologram analysis to confirm synergistic interaction among HER2 and CDK4/6 inhibitors in breast cancers, which serves as a positive control for the identification of synergy by drug-drug 'checkerboard' experiments (*Goel et al., 2016*).

The Bliss model assesses the efficacy of cytotoxic drugs according to the proportion of cells killed (rather than potency, as measured by $IC_{50}$ for example), and drugs are scored as interacting only if their combined effect exceeds a null model of independence involving statistically independent probabilities of cell killing (*Bliss, 1939*). Specifically, if given doses of drugs $a$ or $b$ alone kill proportions of cells equal to $p_a$ or $p_b$ and these probabilities of death are not correlated, then the proportion of cells expected to die from a combination of these drugs at the same doses is $p_{expected} = p_a + (1 - p_a) \times p_b = p_a + p_b - p_a \times p_b$. In this model, synergy or antagonism is defined as an observed excess or deficiency in the proportion of cells killed; we assessed this on a logarithmic scale (Excess over Bliss, $EOB = \text{Log}_{10}(1 - p_{expected}) - \text{Log}_{10}(1 - p_{observed})$) to identify increases in fractional kill. For example, in comparing 99% kill with 99.9% kill, the difference is less than 1% on a linear scale, but a logarithmic scale correctly reveals a 10-fold difference in the probability of survival. By this analysis, we find that pairs of drugs in R-CHOP are largely independent, except that killing by O is strongly antagonized by the presence of either C or H (*Figure 1A*). Antagonism may be a consequence of the effects of these drugs on the cell cycle: killing of mitotic cells by O is expected to decrease when C- or H-induced DNA damage prevents entry into mitosis (*Barlogie et al., 1976*; *Cutts, 1961*; *Davidoff and Mendelow, 1993*).

In isobologram analysis, contour lines (isoboles) corresponding to a constant phenotype (the fraction of cells killed) are plotted across a two-way dose-response landscape (*Greco et al., 1995*; *Loewe, 1953*). The shape of the contours is diagnostic of drug interaction: straight contours

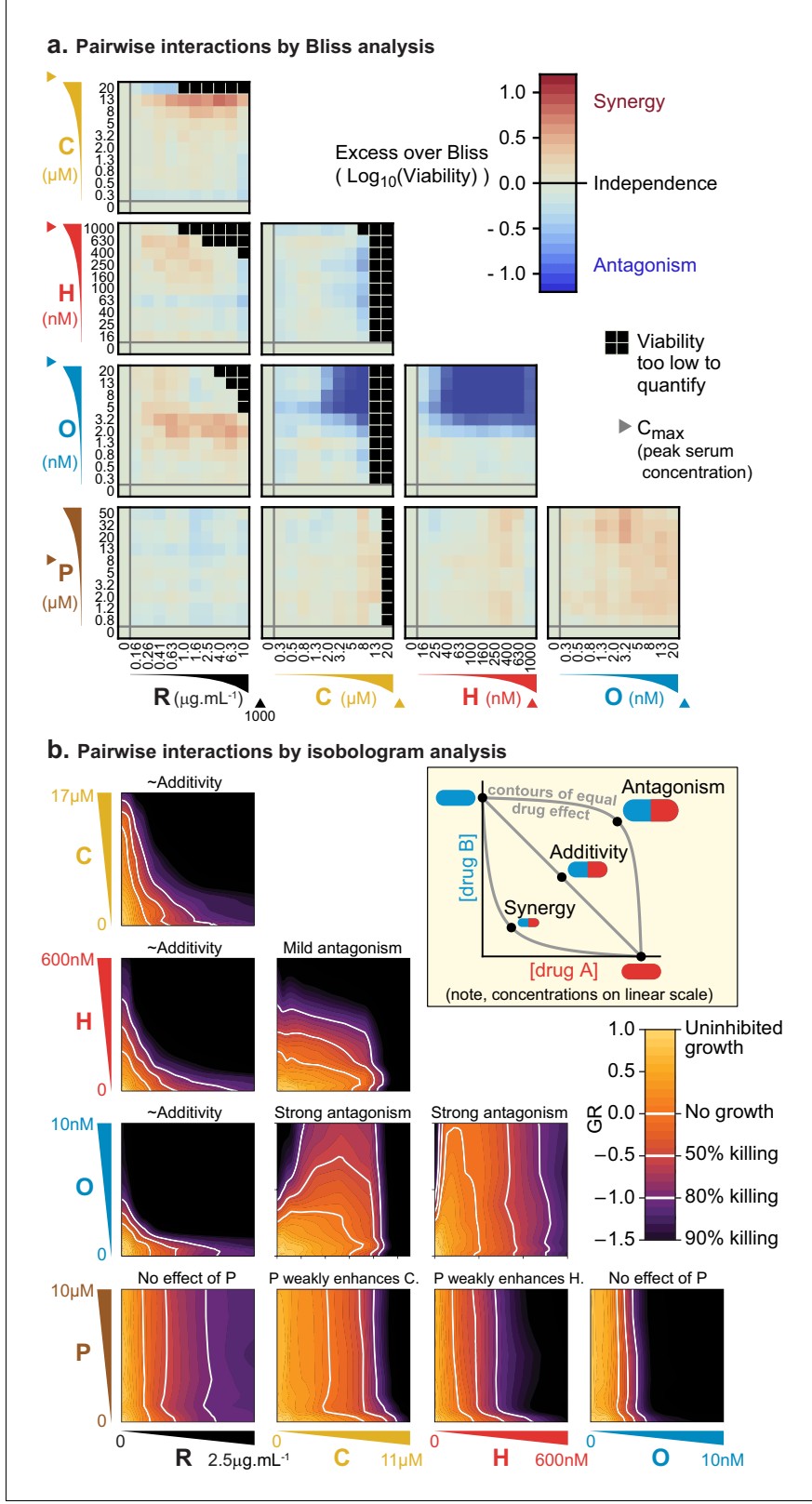

**Figure 1.** Pairs of drugs in R-CHOP exhibit little synergy, but some strong antagonism, in a Diffuse Large B-Cell Lymphoma cell line. (a) Pfeiffer cells grown in microtiter plates were treated with drug combinations for 72 hr followed by a luminescence-based assay for cell viability. 'Excess over Bliss' measures the observed deviation from Bliss Independence. Triangles on axes indicate each drug's approximate $C_{max}$, or peak serum concentration in

*Figure 1 continued on next page*

*Figure 1 continued*

patients. (**b**) Isobologram analysis of the same experiments; luminescence relative to untreated control cells was converted into a GR metric (*Hafner et al., 2016*) to distinguish cytostatic from cytotoxic effects. White contours highlight thresholds equivalent to complete growth inhibition (GR = 0), and complete growth inhibition plus 50% or 80% cell killing (GR <0). Inset: principles of isobologram analysis; isoboles are contours of equal drug effect, which are straight lines in the case of 'additivity'.

The online version of this article includes the following figure supplement(s) for figure 1:

**Figure supplement 1.** Measuring cytotoxic responses to R-CHOP drugs in DLBCL cultures.

correspond to drug additivity, convex contours to synergy and concave contours to antagonism (*Figure 1B*, inset). This arises because straight contours correspond to a scenario of 'dose-equivalence'; that is, a unit of drug *a* can substitute for a unit of drug *b* and *vice versa* (when units are normalized by potency). When contours are convex, a disproportionately small dose of *a* plus *b* is as active as a full dose of either monotherapy. Isobologram analysis of drug pairs in R-CHOP confirmed results from Bliss analysis, namely that interactions among R-CHOP constituents range from strongly antagonistic to approximately additive (*Figure 1B*). As discussed earlier, prednisolone was not cytotoxic on its own but it slightly sensitized cells to C and to H. CMC by rituximab was approximately additive with each of C, H, and O, whereas C and H severely antagonized O. Note that the small convexity visible in *Figure 1B* when R is combined with other agents does not meet the 2-fold deviation from additivity that is the recommended threshold for avoiding false claims of synergy due to errors in measurement (*Odds, 2003*). We conclude that no drug pair in R-CHOP exhibits synergistic interaction by either isobologram analysis (Loewe additivity) or Bliss independence.

To test for higher order interactions, we exposed each of the three different DLBCL cell lines to all 26 possible combinations of 2, 3, 4, or five drugs (*Figure 2A*). Because high-order combinations cannot feasibly be studied across multi-dimensional dose 'checkerboards', R-CHOP constituents were tested at fixed ratios scaled so that constituents were equipotent with respect to cell killing when assayed individually (*Figure 2—figure supplement 1A*). The activity of drug combinations was then quantified by *Fractional Inhibitory Concentrations* (FIC [*Elion et al., 1954*], also known as *Combination Index* [*Chou, 2010*]), which is a fixed-ratio simplification of Loewe's isobologram analysis. If single drugs achieve a given effect magnitude, 50% killing for example, at concentrations $A$, $B$, or $C$ (using three drugs as an example), and their combination achieves the same effect at concentrations $a + b + c$, then $FIC = a/A + b/B + c/C$ (note that Loewe additivity corresponds to FIC = 1 and synergy is commonly defined as FIC <0.5). In all three DLBCL cultures, we observed that small excesses over additivity for R and P on CHO was balanced by antagonism within CHO, producing net effects ranging from approximately additive to slightly antagonistic (for five drugs in Pfeiffer FIC = 0.80 ± 0.15; for SU-DHL-6 FIC = 1.1 ± 0.3 and for SU-DHL-4 FIC = 1.7 ± 0.2; 95% confidence, n = 4–8; *Figure 2B,C*). The absence of synergy across high-order combinations was supported by Bliss analysis of the same data (*Figure 2—figure supplement 1B*). Emergent pharmacological interactions involving combinations of 3 or more drugs can be identified as deviations from the assumption of dose additivity using data from lower order drug interactions (*Cokol et al., 2017*); nearly all such terms supported the hypothesis of no interaction (emergent FIC = 1) with the only substantial deviations representing mild antagonism (emergent FIC up to 1.5) (*Figure 2—figure supplement 1C*). We conclude that R-CHOP does not exhibit significant synergy among its constituent drugs in cell culture.

## DLBCL clones resistant to one drug in R-CHOP rarely resist multiple drugs

To test the hypothesis that low cross-resistance is important for a curative therapy, we asked whether clones resistant to any single drug in R-CHOP remain susceptible to at least one other drug in the combination. DLBCL genomes are relatively complex, possessing a mixture of single nucleotide polymorphisms and copy number gains and losses (*Pasqualucci et al., 2011*; *Sebastián et al., 2016*). We therefore looked for resistance mutations using three complementary approaches: (i) random mutagenesis coupled to clone tracing, (ii) genetic knockdown via CRISPR interference (CRISPRi) for loss of function mutations, and (iii) overexpression via CRISPR activation (CRISPRa) for gain of

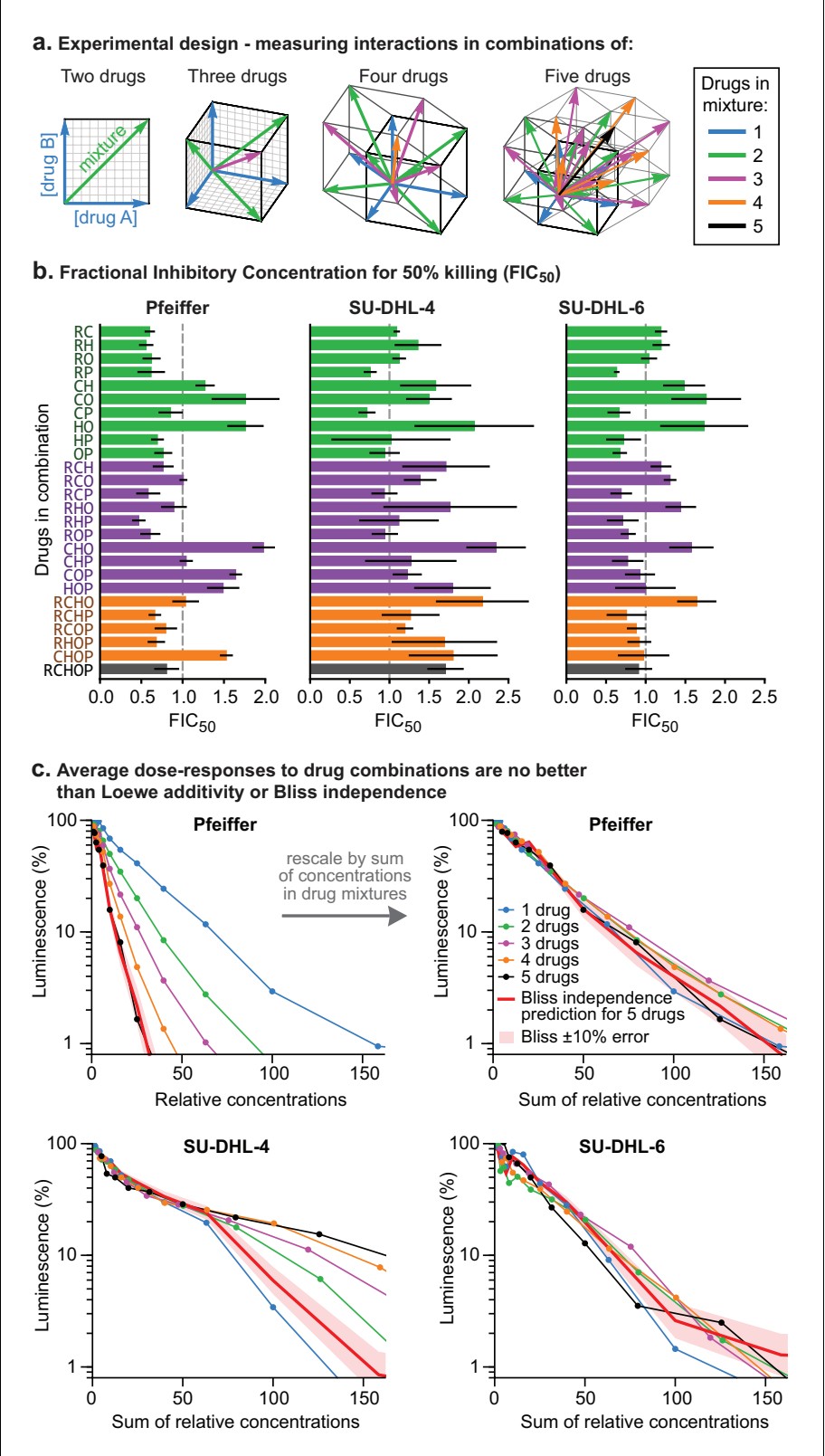

**Figure 2.** Higher order drug combinations do not exhibit synergistic cell killing. (a) Experimental design: two or more drugs were mixed in equipotent ratios such that they similarly contributed to cytotoxicity as the dose of the mixture was increased. Dose gradients of drug mixtures span diagonal lines in multi-drug concentration space. (b) Synergy or antagonism of multidrug combinations was quantified by Fractional Inhibitory Concentrations (FIC) at

*Figure 2 continued on next page*

*Figure 2 continued*

the 50% killing threshold (*Figure 1—figure supplement 1D*). Error bars are 95% confidence intervals (n = 4 per point along dose response). (c) Average dose response functions to single drugs or mixtures of different numbers of drugs (i.e., average of single-drug responses; average of drug pair responses, etc.). Red line: expected response to R-CHOP drugs according to the Bliss Independence model; pale red area:±10% error in number of log-kills around the Bliss Independence model. Top left: Horizontal axis shows the amount of each drug present in a mixture (units are scaled to align single-agent activity; *Figure 2—figure supplement 1A*). Top right, bottom left, bottom right: Horizontal axis is the sum of drug concentrations.

The online version of this article includes the following figure supplement(s) for figure 2:

**Figure supplement 1.** Measuring high-order interactions among R-CHOP drugs with equipotent combinations.

function mutations (*Figure 3A*). In a hypothetical multi-drug treatment, it is not possible to distinguish between single-drug or multi-drug resistance as either could increase the survival of a mutated clone (*Figure 3B*). A further complication is that strongly antagonistic drug combinations, such as C, H, and O, can select for *sensitivity* to the antagonizing agent (*Chait et al., 2007*). We therefore scored mutations as conferring true cross-resistance by applying drugs individually and identifying mutant cells significantly enriched in two or more conditions (*Figure 3C*). This was accomplished by generating a pool of mutagenized/CRISPR-transformed cells in which each cell carried a unique DNA barcode (or single guide RNA (sgRNA) that also acts as a barcode). Cells were split into independent cultures and then treated with a single component of R-CHOP. The abundance of DNA barcodes in each culture was measured before and after drug exposure by high-throughput DNA sequencing followed by enrichment analysis.

For random mutagenesis and clone tracing, Pfeiffer cells were mutagenized with N-ethyl-N-nitrosourea (ENU), which induces point mutations and chromosome aberrations (*Sanger and Eisen, 1976*; *Shibuya and Morimoto, 1993*). One million mutagenized clones were barcoded using a lentiviral DNA barcode library (ClonTracer; *Bhang et al., 2015*). Because the library was complex ($\approx 7 \times 10^7$ barcodes) and infection performed at low multiplicity (MOI ~0.1), over 99% of barcoded clones are expected to contain a unique barcode. Barcoded cells were expanded in puromycin to select for the lentiviral vector. From a single well-mixed suspension of cells, a batch was reserved to measure pre-treatment barcode frequencies, and the remainder was distributed into 18 replicate cultures (three per drug tested) with each culture providing 12-fold coverage of barcoded clones (*Figure 3—figure supplement 1A*).

To model the clinical scenario of strong selection pressure from intensive treatment cycles (as opposed to continuous low dose therapy), drugs were applied for 72 hr at a dose established in a pilot study as the highest dose allowing any surviving cells to re-grow in drug-free media in under 2 weeks (Materials and methods). Cultures were exposed to two rounds of drug treatment followed by a recovery period of 4 to 11 days as needed (*Figure 3—figure supplement 1A*). Because prednisolone monotherapy only slowed growth, which is difficult to score in a short duration culture, cells were treated with prednisolone at 20 µM for 20 days (the R-CHOP regimen contains multiple five-day courses of prednisone). Enrichment for specific clones was calculated based on relative barcode frequencies prior to and after treatment.

Thousands of clones were reproducibly enriched in replicate cultures exposed to the same drug. To score cross-resistance and account for culture-to-culture variation across repeats, we constructed an error model by scrambling barcode identities within each replicate. This revealed that at least 300-fold more barcodes were $\geq$10 fold enriched in repeat experiments for any single drug than expected by chance (*Figure 4A*). We also accounted for fitness differences observed in vehicle-only cultures (~1% of barcodes were enriched $\geq$10 fold in the presence of DMSO; see Materials and methods). Correlations between enrichment scores in replicate drug treatments were highly significant ($p<10^{-900}$; n $\approx$ $10^6$) although of modest magnitude (0.1 to 0.3) (*Figure 4B*). This arises because drug exposure imposes population bottlenecks on non-resistant clones, which represent the majority of the population, causing barcodes to be detected, or not, on a stochastic basis. Among barcodes with non-zero counts in replicate experiments, correlation was higher (0.35 to 0.58). We used the geometric mean of enrichment for each barcode as a metric of drug resistance across replicates. Instances of stochastic (and thus irreproducible) enrichment are strongly penalized by this metric; conversely, barcodes are favored if they are reproducibly enriched in independent

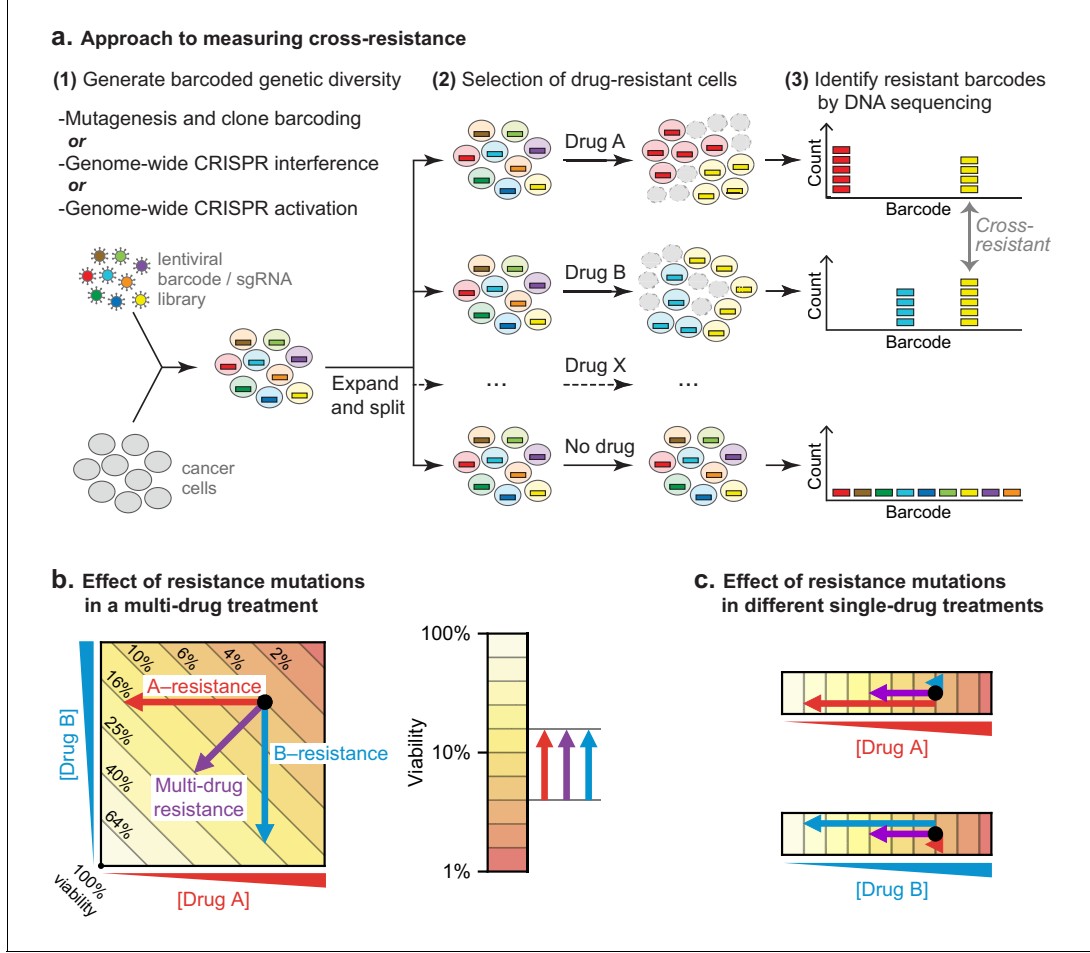

**Figure 3.** Strategy for measuring cross-resistance between drugs. (**a**) Cells were mutagenized and barcoded using one of three approaches: (i) random mutagenesis and clone tracing, (ii) knockdown by CRISPRi or (iii) overexpression by CRISPRa. $10^6$ mutagenized clones or genome-wide CRISPRi/a libraries were expanded and split into replicate cultures, treated with single drugs, and DNA barcodes/sgRNAs abundance was measured by DNA sequencing. The resistance of cells to drug treatment was scored based on the degree of barcode enrichment, and cross-resistance was determined by significant enrichment in two or more drug treatments. (**b**) Schematic showing importance of selecting for resistance to single drugs not cocktails. Arrows: resistance is analogous to lower drug concentration and moves cells to different coordinates; cross-resistance (purple arrow) has same net effect as more penetrant single-drug resistance mutations (red, blue arrows). (**c**) By selecting mutations on single drugs the magnitude of the effect on each drug is known.

The online version of this article includes the following figure supplement(s) for figure 3:

**Figure supplement 1.** Dosing schedule and replication strategy for all three approaches taken to isolate cells resistant to single cytotoxic drugs in the R-CHOP combination.

cultures (which is evidence of heritability). Drug resistant clones could potentially be lost in any one replicate due to stochastic drift, causing an underestimate of drug resistance. However, our use of large initial populations, triplicate experiments, and a metric that scores as positive barcode enrichment in 2 of 3 replicates help to minimize this concern.

The error model constructed from scrambled barcodes was used to estimate the false discovery rate for barcode enrichment. We found that the stronger the geometric mean enrichment, the less likely it was for enrichment to occur randomly (*Figure 4C*). False discovery of coincident enrichment exceeding 10-fold in two or more drugs was rare (<2.5%) and we therefore selected this threshold for subsequent analysis. For each of the four individually active drugs (i.e. RCHO), 2000 to 13,000 barcodes were identified with geometric mean enrichment ≥10 fold, representing resistance frequencies of $2 \times 10^{-3}$ to $1 \times 10^{-2}$. The vast majority of enriched clones were unique to one drug,

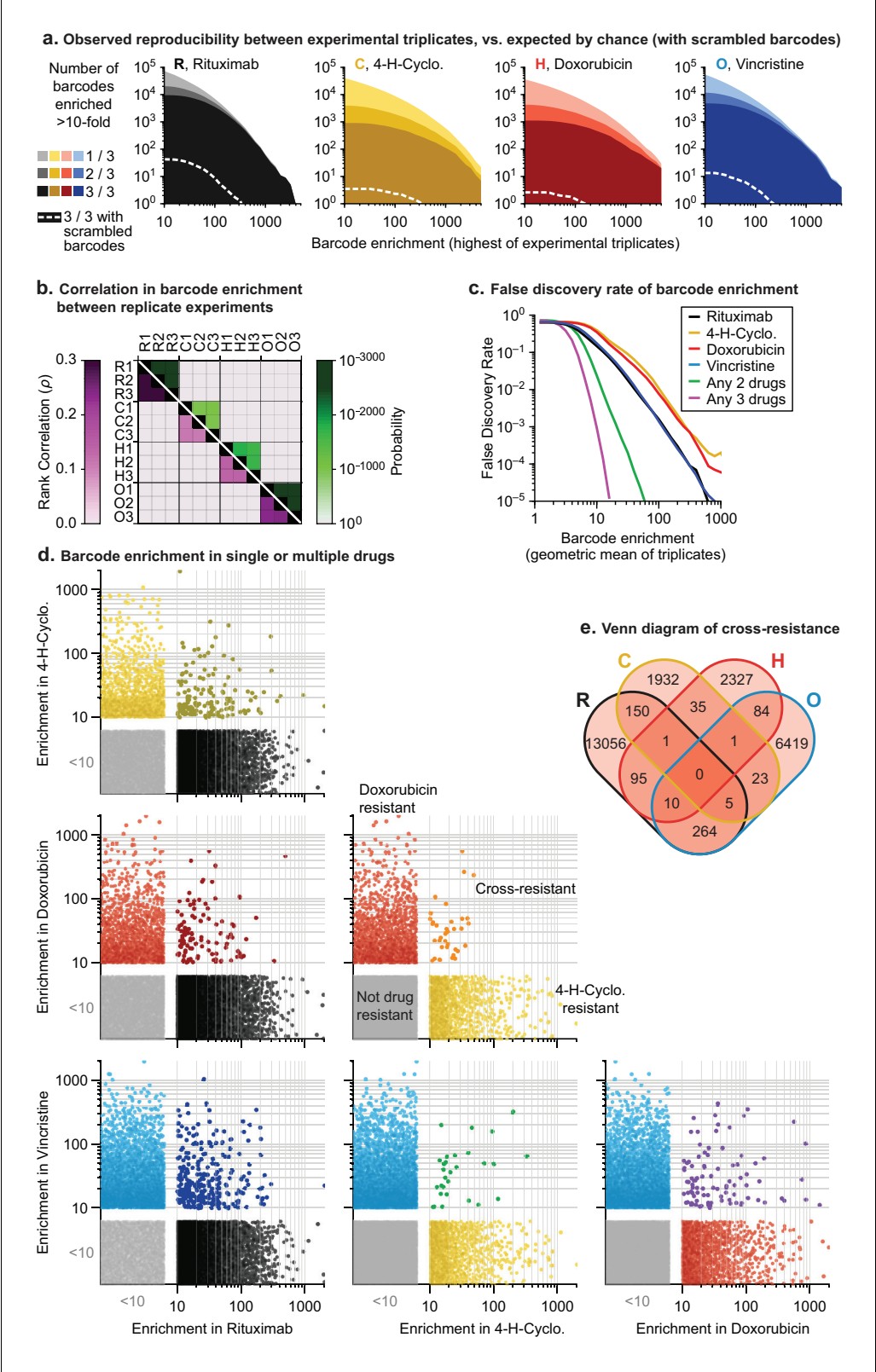

**Figure 4.** In mutagenized clones single-drug resistance is common but multi-drug resistance is rare. (a) Reproducibility of DNA barcode enrichment among triplicate drug treatments. Horizontal axis is the highest value for each barcode's enrichment scores in any replicate. Vertical axis is the number of barcodes for which 1, 2, or three triplicates had enrichment ≥10. Dashed white line: error model indicating repeated enrichment expected by random chance; see Materials and methods. (b) Matrix of Spearman rank correlation in enrichment scores between drug treatments (ρ in pink scale;

*Figure 4 continued on next page*

*Figure 4 continued*

one-sided p-value in green scale). (**c**) False discovery rate of barcode enrichment, per magnitude of enrichment (geometric mean of triplicates), was computed by comparing observed barcode enrichment to an error model of scrambled barcodes ($10^{10}$ triplicates simulated by scrambling actual data). At geometric mean enrichment = 10, false discovery rate for 2-drug and 3-drug resistance is 2.5% and 0.1%, respectively. (**d**) Scatterplots of barcode enrichment scores (geometric mean of biological triplicates for each drug) for each pair of two drugs in RCHO. Each dot represents a single barcode. Enrichment scores < 10 are deemed not significant. (**e**) Venn diagram of the number of barcodes exhibiting resistance (geometric mean enrichment ≥10) to single or multiple drugs.

The online version of this article includes the following source data and figure supplement(s) for figure 4:

**Source data 1.** Barcode counts for all clone tracing experiments.
**Figure supplement 1.** Prednisolone-resistant clones show low cross-resistance to other drugs in R-CHOP, and repeats of vincristine show high cross-resistance.

with only 30 to 300 clones (depending on the pair of drugs) enriched in two different conditions; the latter is diagnostic of double-drug resistance (*Figure 4D,E*). Triple-drug resistant clones were even less abundant (between 1 and 10 clones per set of three drugs) and no clones were identified that conferred resistance to each of R, C, H and O when applied individually (*Figure 4E*). Note that these frequencies are based on determining the co-occurrence of barcodes enriched in single-drug experiments. As described in *Figure 3B*, barcode enrichment in an experiment that applied four or five drugs at the same time cannot be expected to distinguish between resistance to some drugs or all drugs; in addition, the likelihood that a clone that is multiply drug resistant might be missed by stochastic drift increases. Clones enriched by prolonged exposure to prednisolone at concentrations that slowed growth also exhibited low overlap with barcodes enriched for other constituents of R-CHOP (*Figure 4—figure supplement 1A,B*).

## CRISPRi/a screening identifies diverse mechanisms of drug resistance

Screening genome-wide sgRNA libraries using CRISPR-Cas9 has the advantage that it yields the identities of genes conferring drug resistance as opposed to barcodes for unknown ENU-mutated loci. CRISPRi screening identifies loss of function resistance mutations and was performed in Pfeiffer cells by expressing nuclease-dead Cas9 fused to the transcriptional repression domain KRAB (dCas9-KRAB) (*Gilbert et al., 2013*). CRISPRa screening identifies overexpression resistance mutations and was performed in cells by co-expressing dCas9 fused to SunTag (a repeating peptide array) and a SunTag-binding antibody fused to the VP64 transcriptional activator (*Tanenbaum et al., 2014*). This approach requires clonal selection of a co-expressing cell line in which the ratio of dCas9:VP64 is fixed; otherwise, cell-to-cell variability complicates screening for overexpression phenotypes. However, we were unable to generate monoclonal lineages of Pfeiffer cells expressing dCas9 and VP64. In other DLBCL cell lines, lentiviral transduction was inefficient (a known property of B lymphocytes and lymphomas [*Li et al., 2001*]). We therefore performed CRISPRa screens in the chronic myeloid leukemia (CML) cell line K562, which can be efficiently transduced and cloned. For CRISPRi in Pfeiffer cells it was possible to screen for resistance to four drugs (R, C, H and O) but for CRISPRa in K562 cells, screening was possible only for C, H and O. Of note, these drugs have been used historically in the treatment of CML, and we validated (below) that screen hits identified in K562 CML cells could be reproduced in Pfeiffer DLBCL cells.

We used RT-qPCR to confirm that transduction of sgRNAs in cells expressing the appropriate dCas9 fusion protein caused strong repression of a set of test target genes by CRISPRi and strong activation by CRISPRa (*Figure 5—figure supplement 1A*). We then used lentivirus at low multiplicity (MOI ≤0.4) to infect CRISPRi and CRISPRa-expressing cells with second generation genome-scale sgRNA libraries, which are highly active by virtue of having optimized target sites that account for nucleosome positioning (*Horlbeck et al., 2016*). Both libraries contain 10 sgRNAs per gene, and approximately 4000 control sgRNAs designed to have no target. Following expansion, infected cells were exposed to drug (or vehicle) for two to three 72 hr drug pulses separated by recovery periods of up to 5 days as needed (*Figure 3—figure supplement 1B,C*). Hits were identified by sequencing sgRNAs (*Figure 5—source data 1*). The impact of each sgRNA on drug sensitivity was quantified by the 'rho phenotype' (*Kampmann et al., 2013*), which is one in the case of complete resistance, 0 in the case of sensitivity matching the parental cell line (as determined using non-targeting control sgRNAs), and <0 for hypersensitivity (Materials and methods; *Figure 5—source data 2*). Across 10

sgRNAs for each gene we calculated the mean of the strongest five rho phenotypes by absolute value, and the p-value of all 10 rho phenotypes as compared to the 4000 control sgRNAs (Mann-Whitney test) (Gilbert et al., 2013). Random permutations of 10 control sgRNAs were assembled to create ≈ 19,000 'negative control genes', matching the number of real gene targets and with phenotypes specific to each drug screen. For all drugs tested, plots of gene phenotype vs. significance ('volcano plots') revealed many gene perturbations conferring drug resistance or hypersensitivity (*Figure 5* and *Figure 5—source data 3*).

Hits from CRISPRi and CRISPRa were consistent with known mechanisms of drug action: knockdown of direct targets was observed to confer resistance to rituximab (*MS4A1* encoding CD20) and doxorubicin (*TOP2A* encoding topoisomerase II) (*Thorn et al., 2011*; *Weiner, 2010*) whereas overexpression of *TUBB* (which encodes β-tubulin) conferred resistance to vincristine. Cyclophosphamide functions by inducing interstrand crosslinks in genomic DNA via alkylation. CRISPRi identified multiple genes involved in the DNA damage response: cyclophosphamide resistance was conferred, for example, by knockdown of *SLFN11* which blocks progression of stressed replication forks (*Murai et al., 2018*; *Zoppoli et al., 2012*) and hypersensitivity (measured in a supplemental screen at a lower cyclophosphamide dose; *Figure 5—figure supplement 1B,C*) was caused by knockdown of genes involved in DNA interstrand crosslink repair (e.g. *FANCE*, *FANCD2*, *UBE2T*, *FANCI*, *ATRIP*) and double-strand break repair (e.g. *BRIP1*, *BARD1*, *BRCA1*, *BRCA2*). The therapeutic window for cyclophosphamide arises from tissue-specific expression of aldehyde dehydrogenases (ALDHs) which are the primary enzymes involved in cyclophosphamide inactivation (*Cox et al., 1975*); overexpression of *ALDH1A1* and *ALDH1B1* as well as aldo-keto reductases (AKRs) that metabolize cytotoxic products of cyclophosphamide (*Penning, 2017*) all conferred resistance in our screen. Detailed study of these genes is beyond the scope of this manuscript (full results are in *Figure 5—source data 3*) but from these data we conclude that CRISPRi/a screening successfully identifies biologically relevant genes involved in resistance to RCHO.

To test the robustness of results from whole-genome screens, we performed individual validation studies with selected sgRNAs. We constructed single knockdown or overexpression cell lines for each of nine CRISPRi and eight CRISPRa sgRNAs conferring single or multi-drug resistance phenotypes, and measured their drug sensitivity in dose-response experiments (for a total of 9 genes × 4 drugs=36 validation experiments for CRISPRi; 8 × 3 = 24 for CRISPRa). The $IC_{50}$ values for drug responses in Pfeiffer cells as measured in CRISPRi validation experiments were strongly correlated with resistance phenotypes from the original knockdown screen (r = 0.66, p<$10^{-5}$; n = 36 gene–drug interactions; *Figure 6A* and *Figure 6—figure supplement 1A*) as were $IC_{50}$ values for CRISPRa validation studies in K562 CML cells (r = 0.84, p<$10^{-5}$; *Figure 6B* and *Figure 6—figure supplement 1B*). To test if resistance genes identified in K562 cells have similar phenotypes in DLBCL cells, we generated a polyclonal CRISPRa Pfeiffer cell culture and derived individual overexpression mutants by transduction of single sgRNAs. Gene overexpression is less efficient in this setting than in K562 cells (*Figure 6—figure supplement 1B*) but we nonetheless found that CRISPRa resulted in changes in $IC_{50}$ values in Pfeiffer cells that were strongly correlated with changes observed in K562 validation experiments (r = 0.82, p<$10^{-5}$; n = 24 gene-drug interactions) and with resistance phenotypes obtained in the overexpression screen in K562 (r = 0.62, p=0.001; *Figure 6—figure supplement 1C, D*).

To measure overexpression-mediated drug resistance with greater sensitivity, we also performed competition assays by mixing two Pfeiffer cultures, one expressing a gene-targeting sgRNA from the validation studies described above and the second a non-targeting sgRNA. We then measured the change in the ratio of cultures by qPCR following two cycles of drug treatment and recovery (in the same manner as for genome-wide screens; *Figure 3—figure supplement 1*). These experiments demonstrated a strong correlation between sgRNA-induced competitive fitness in DLBCL cells grown in the presence of drug and drug resistance as measured in the overexpression screens (r = 0.69, p=0.0002; *Figure 6C*). We conclude that genome-wide knockdown and overexpression screens yielded robust and reproducible hits, and that overexpression-mediated resistance identified in K562 cells is largely recapitulated in DLBCL cells.

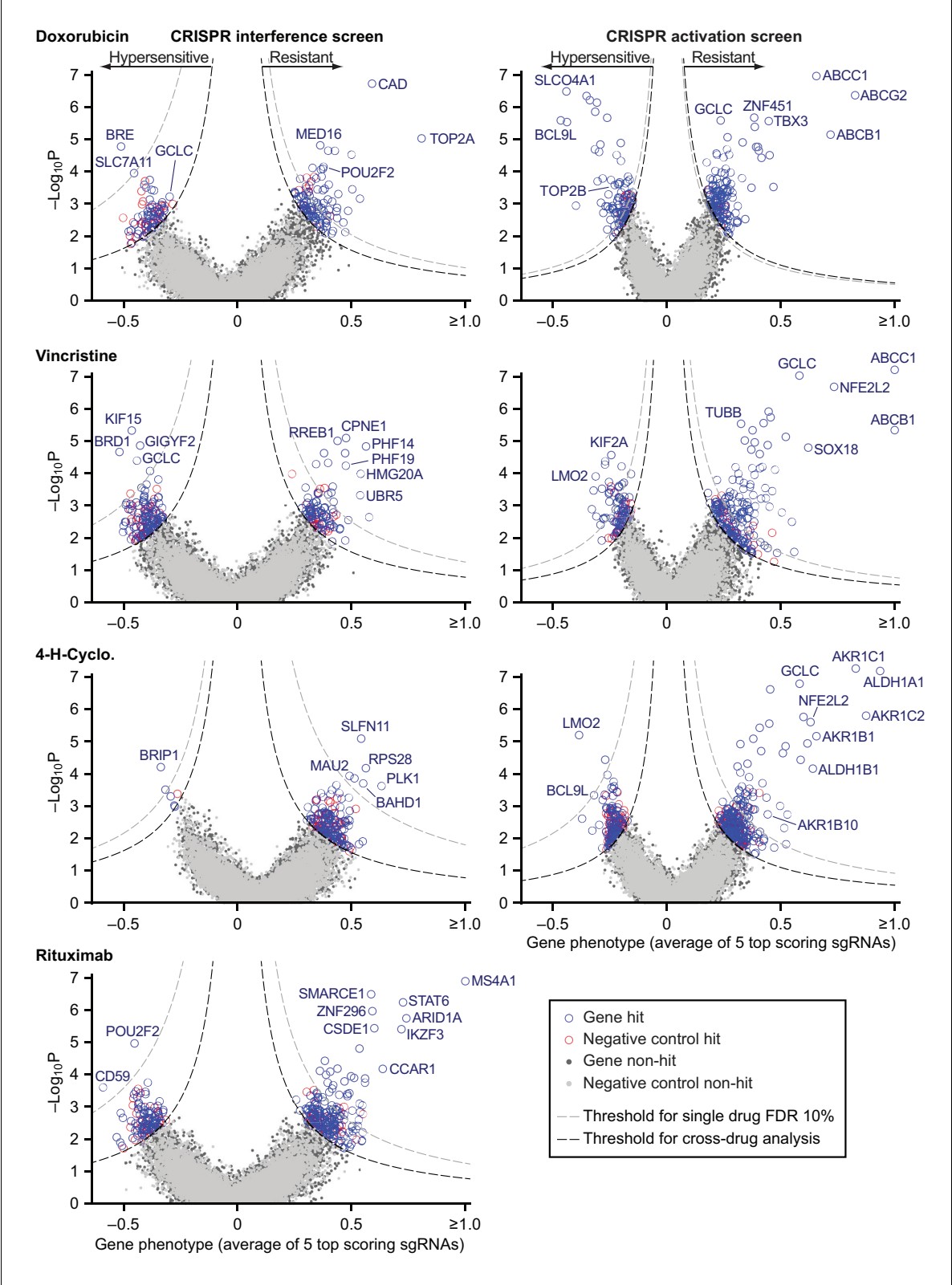

**Figure 5.** Identification of mechanisms of single drug resistance by genome-wide CRISPRi and CRISPRa screening. Volcano plots of gene phenotype and p-value for CRISPRi (left) and CRISPRa (right) screens of single R-CHOP drugs. Phenotype of 1 is full resistance, 0 is parental sensitivity, <0 is hypersensitivity. The coordinate of each gene was determined by the average phenotype of the five most active sgRNAs targeting that gene and -log$_{10}$ of the p-value (Mann-Whitney test of phenotypes for all targeting sgRNAs compared to 4000 non-targeting controls). Negative control genes were

*Figure 5 continued on next page*

*Figure 5 continued*

generated by randomly grouping sets of non-targeting sgRNAs. Gray dashed line: threshold for 10% FDR for single-drug resistance, or hypersensitivity. Black dashed line: threshold for cross-resistance set to yield less than one double-resistant negative control gene out of all possible drug pairs (equal to multi-drug resistance FDR 4% for CRISPRi and 2% for CRISPRa). Labeled genes are a partial list of top scoring hits.

The online version of this article includes the following source data and figure supplement(s) for figure 5:

**Source data 1.** sgRNA counts for all CRISPR screens.
**Source data 2.** sgRNA phenotype scores for all CRISPR screens.
**Source data 3.** Gene scores for all CRISPR screens.
**Figure supplement 1.** CRISPRi/a cell lines strongly alter gene expression of targeted genes and additional cyclophosphamide CRISPRi screen identifies hypersensitive hits in the DNA interstrand crosslink pathway.

## Knockdown and overexpression mutations identified by CRISPRi/a do not confer pan-drug resistance

Next, we asked whether any of the gene perturbations identified by knockdown or overexpression screening conferred resistance to multiple drugs. For each screen, we calculated a single resistance score that takes into account both effect size and the significance of enrichment (mean rho phenotype $\times$ -log$_{10}$P). We selected a cut-off in resistance scores that yielded less than one false-positive example of multi-drug resistance per $\approx$19,000 negative control genes (the number of real gene targets). This cut-off is lenient in scoring for single-drug resistance because it is designed to reduce the chance that true cross-resistance will be missed. It therefore ensures a *more* stringent test of Law and Frei's hypothesis (*Figure 7—figure supplement 1A,C*). CRISPRi yielded 19 genes whose knockdown conferred resistance to two drugs, and four genes conferring resistance to three drugs (*Figure 7A,B* and *Figure 7—figure supplement 1B*). For example, resistance to rituximab and doxorubicin was conferred by CRISPRi of *SMARCE1*, a known tumor suppressor in DLBCL and other cancers (*Shain and Pollack, 2013*), and by CRISPRi of *CAD*, a protein involved in pyrimidine biosynthesis whose knockdown causes S phase arrest (*Jost et al., 2017*). Genes that conferred triple resistance when knocked down were involved in translation initiation, chromatin modification, protein degradation and the mediator complex; these gene knockdowns conferred mild resistance as compared to those producing single and double resistance (resistance score <2) and also reduced cell proliferation in the absence of drug (p-value for growth defect <10$^{-5}$; *Figure 7—source data 1*). Thus, these multi-drug resistance genes may act by reducing rates of proliferation, a phenotype that generally predisposes cells to chemotherapy resistance (as reported for *CAD*; *Jost et al., 2017*). No genes were identified by CRISPRi whose knockdown conferred resistance to every drug in RCHO.

Screening by CRISPRa identified 42 genes whose overexpression conferred resistance to two drugs and four genes that conferred resistance to three drugs (*Figure 7C,D* and *Figure 7—figure supplement 1D*). Overexpression of the *ABCB1* and *ABCC1* ATP-binding cassette (ABC) transporters resulted in resistance to H and O, but not to C (*Figure 7C*), and overexpression of the *ABCG2* ABC transporter conferred resistance to H alone (*Figure 5*); upregulation of drug export via overexpression of ABC transporters has been implicated in resistance to many drugs (*Choi, 2005*). Two of four genes whose activation conferred triple-drug resistance (to C, H and O) were linked to glutathione biosynthesis: *GCLC*, which catalyzes the first step in glutathione production, and *NFE2L2*, a transcription factor for *GCLC* and other genes involved in response to xenobiotics and oxidative stress (*Figure 7C*) (*Kitamura and Motohashi, 2018*; *Zanotto-Filho et al., 2016*). Glutathione plays an important role in resistance to chemotherapy (*Bansal and Simon, 2018*), and high expression of glutathione family genes is strongly associated with poor overall survival in DLBCL on CHOP (*Andreadis et al., 2007*). Further supporting the importance of glutathione for chemotherapy responsiveness, knockdown of *GCLC* conferred hypersensitivity to H and O, and knockdown of the main transporter of cystine (which is limiting for glutathione synthesis), *SLC7A11*, conferred hypersensitivity to H (*Figure 5*). Thus, CRISPRa identified multiple genes associated with previously described or suspected mechanisms of drug resistance, but even genes associated with 'multi-drug resistance' such as ABC transporters were observed to confer resistance to only a subset of drugs.

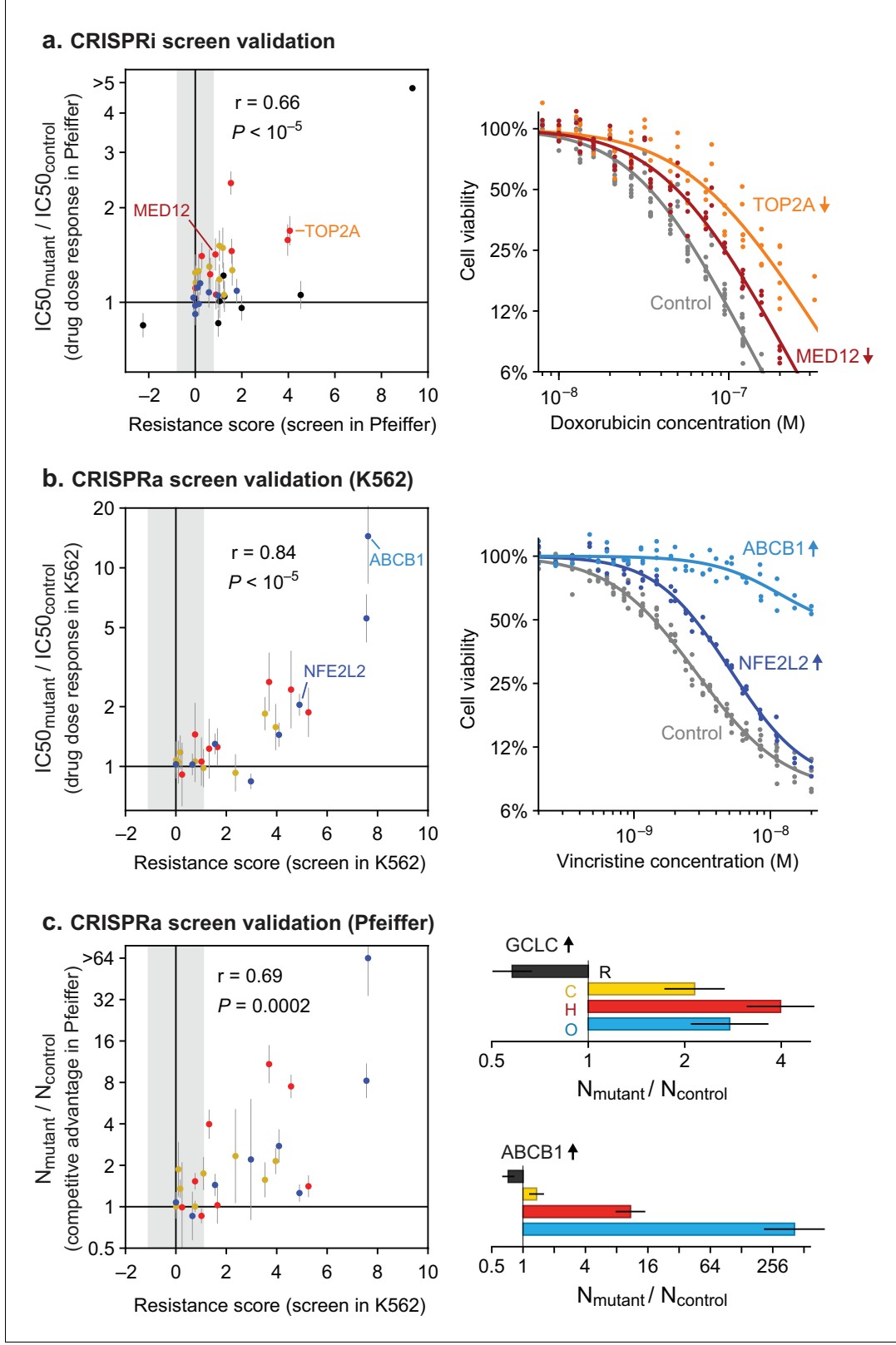

**Figure 6.** Validation of CRISPRi and CRISPRa screen results by individual drug sensitivity measurements. (a) Gene knockdown by CRISPRi produces changes in drug sensitivity (IC50) that are correlated with resistance phenotypes from the genome-wide CRISPRi screen (Pearson correlation r = 0.66, p<10[-5]). Drug dose responses were measured in Pfeiffer CRISPRi cells bearing single sgRNAs, for each of 9 knockdown screen hits, or control non-

*Figure 6 continued on next page*

*Figure 6 continued*

targeting sgRNA. Error bars are 95% confidence intervals in IC50 (determined from curve fit; n = 6). Point color indicates drug: black, rituximab; yellow, 4-hydroperoxy-cyclophosphamide; red, doxorubicin; blue, vincristine. Gray region: threshold in resistance score that was used to identify screen hits. Right: example dose response measurements for control sgRNAs, or sgRNAs inducing *TOP2A* and *MED12* knockdown and consequent doxorubicin resistance. (b) Gene overexpression by CRISPRa produces changes in drug sensitivity (IC50) that are correlated with resistance phenotypes from the genome-wide CRISPRa screen (r = 0.84, p<10$^{-5}$). Drug dose responses (n = 4) were measured in K562 CRISPRa cells bearing sgRNAs for each of 8 overexpression screen hits, or control non-targeting sgRNA. Right: example dose response measurements for control sgRNAs, or sgRNAs inducing overexpression of *ABCB1* and *NFE2L2* and consequent vincristine resistance. (c) Gene overexpression by CRISPRa in DLBCL cells (Pfeiffer) produces drug resistance that is correlated with resistance scores from the CRISPRa screen in K562 cells (r = 0.69, p=0.0002). Pfeiffer CRISPRa cells bearing targeted sgRNAs were mixed at 1:1 ratio with cells bearing non-targeting sgRNA, the co-culture was subjected to two 72 hr drug treatment and recovery periods, and changes in the ratio of mutant to control cells was measured by qPCR of sgRNAs. Error bars are 95% confidence intervals (n = 3). Right: Change in ratio of cells bearing sgRNA that induces *GCLC* overexpression (or *ABCB1* overexpression) versus cells bearing control sgRNA, after drug treatment.

The online version of this article includes the following source data and figure supplement(s) for figure 6:

**Source data 1.** Data obtained in the CRISPR screen validation experiments.
**Figure supplement 1.** Changing transcript abundance with CRISPRi and CRISPRa.

## Cross-resistance between drugs in R-CHOP is close to a theoretical minimum

The degree to which two drugs are subject to shared or distinct resistance mechanisms is expected to vary depending on the drugs, and can be described by a cross-resistance parameter ξ, where $0 \leq$ ξ $\leq 1$. As *Law (1952)* described, if one cell in 10$^A$ has resistance to drug *a*, and one cell in 10$^B$ has resistance to drug *b*, then at least one cell in 10$^{A+B}$ will be resistant to both drugs by chance; this theoretical minimum corresponds to ξ = 0. The largest possible frequency of cross-resistance is the smaller of the single-drug resistance frequencies, which corresponds to ξ = 1. Any observed frequency of cross-resistance can be quantified as a weighted sum of the minimum and maximum possibilities to give a value of ξ between 0 and 1 (Materials and methods). To estimate ξ in clone-tracing studies on ENU-mutagenized cells, we first performed two independent sets of clone tracing experiments (each in triplicate) for resistance to O alone. Perfect replicates should result in ξ = 1; we compared different concentrations of O to mimic differences between drugs in rates of killing, and observed ξ = 0.69 (*Figure 4—figure supplement 1C,D*). Next, examining all combinations of 2, 3 or four drugs we observed uniformly low values for ξ, with an average of ξ = 0.016 (*Figure 8A,B*). This shows that although we observed substantial co-occurrence of clones across drugs, they do not all represent truly cross-resistant mutations, because their frequency can be largely accounted for by the independent acquisition of multiple mutations that each confer resistance to a single drug. Thus, we observed that randomly mutagenized cells exhibited nearly the theoretical minimum rate of cross-resistance, with an absolute frequency of resistance to all drugs in R-CHOP <10$^{-6}$.

In CRISPR screens single gene perturbations are analyzed and only true cross-resistance is detected. Law's prediction can still be applied: if resistance to drugs *a* and *b* is conferred by a fraction of CRISPR perturbations (at frequencies 10$^{-A}$ and 10$^{-B}$) with ξ = 0, perturbations conferring resistance may coincidentally overlap at a frequency of 10$^{-(A+B)}$. In CRISPRi and CRISPRa screens, rates of multi-drug resistance exceeded this minimum, largely due to genes such as transporters whose function is protection against multiple xenobiotics. For example in CRISPRa screens, the theoretical minimum number of 2-drug resistant genes is predicted to be 1 (ξ = 0) and the maximum number 132 (ξ = 1); the observed average for all drug pairs was 18 (ξ = 0.13) (*Figure 8A,B*). Considered together, CRISPRi and CRISPRa screens exhibited an average cross-resistance value of ξ = 0.05. We therefore conclude that multi-drug resistance to the drugs making up R-CHOP is close to its minimum predicted value.

## Other applications: hypersensitivity as a guide to vulnerabilities

The identification of genes involved in drug hypersensitivity has the potential to uncover interactions causing new druggable vulnerabilities. For example, a mutation that confers resistance to drug A

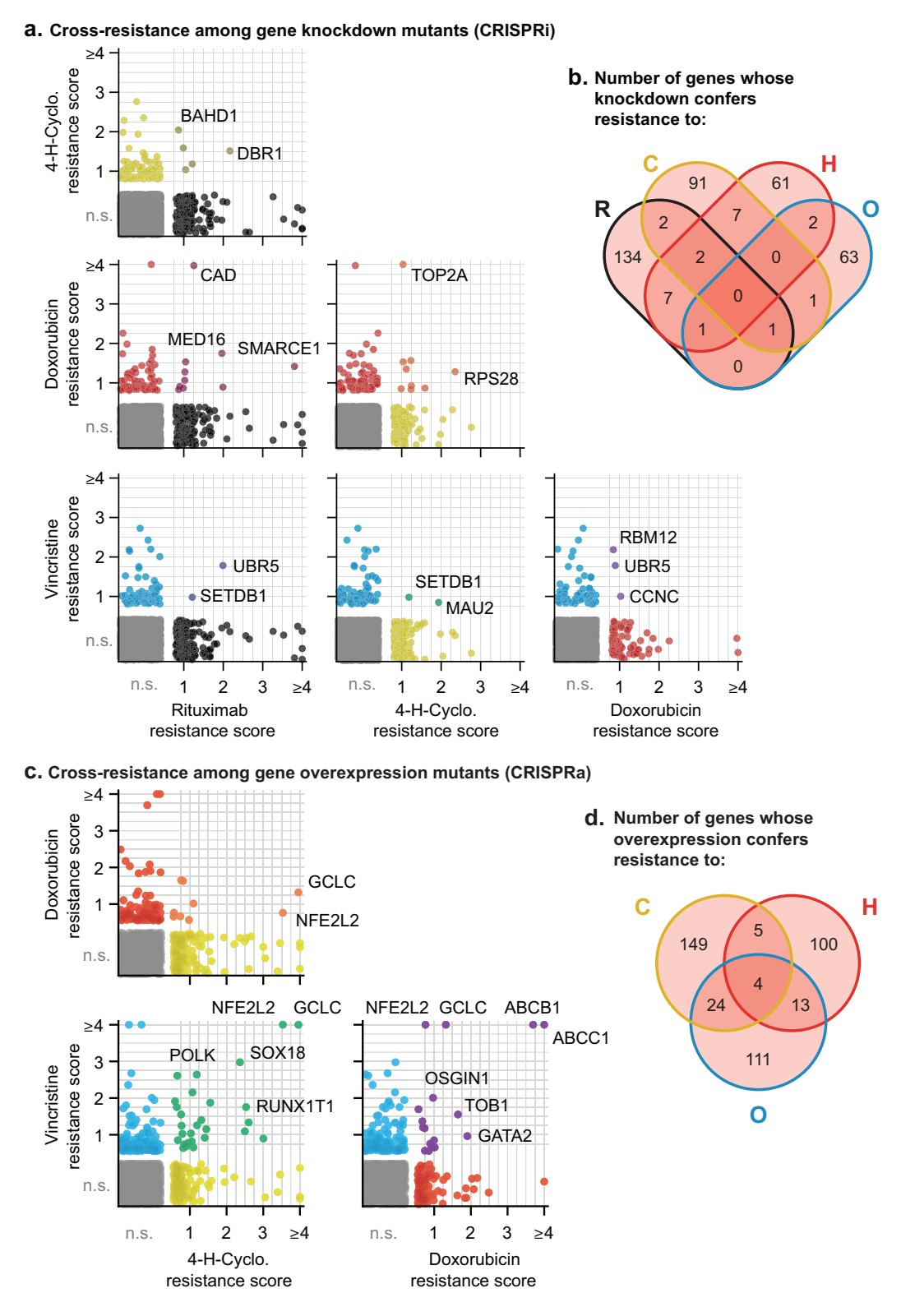

**Figure 7.** Cross-resistance analysis of the CRISPRi and CRISPRa screens reveals a small number of multi-drug resistance mechanisms. (**a**) Scatter plots of resistance scores obtained in CRISPRi screens for each pair of drugs in RCHO; each dot represents a gene. Resistance scores were calculated from the product of the gene phenotype and the significance of the enrichment (-log$_{10}$P). n.s., not significant; genes significant in one drug treatment but not in another are displayed against the left or bottom axis. Labeled genes are a partial list of top scoring hits. (**b**) Venn diagram of the number of genes

*Figure 7 continued on next page*

*Figure 7 continued*

whose knockdown confers resistance to one or multiple drugs in RCHO. (**c**) Scatter plots of resistance scores obtained in CRISPRa screens for each pair of drugs in CHO. Data were analyzed and displayed as in (a). (**d**) Venn diagram of the number of genes whose overexpression confers resistance to one or multiple drugs in CHO.

The online version of this article includes the following source data and figure supplement(s) for figure 7:

**Source data 1.** Gamma growth scores for triple-resistant genes identified in CRISPRi screens.

**Figure supplement 1.** Determination of a cutoff threshold for cross-resistance analysis and identity of all cross-resistant genes for CRISPRi and CRISPRa screens.

**Figure supplement 2.** Determination of a cutoff threshold for cross-hypersensitivity analysis and identity of all cross-hypersensitive genes for CRISPRi and CRISPRa screens.

might confer sensitivity to drug B. Such 'collateral sensitivity' has been extensively studied in the past and remains a relevant concept (*Hutchison, 1965*; *Zhao et al., 2016*). We analyzed drug hypersensitivity in the same manner as resistance (Materials and methods, *Figure 7—figure supplement 2A,C*) and found that, among 778 CRISPRi or CRISPRa resistance genes, only 13 (1.7%) exhibited hypersensitivity to a different drug. Thus, our data indicate that collateral sensitivity does not play a major role in R-CHOP therapy, and resistance is primarily eradicated by drugs that are active against the parental cell population. Collateral sensitivities may be relevant to other drug combinations and

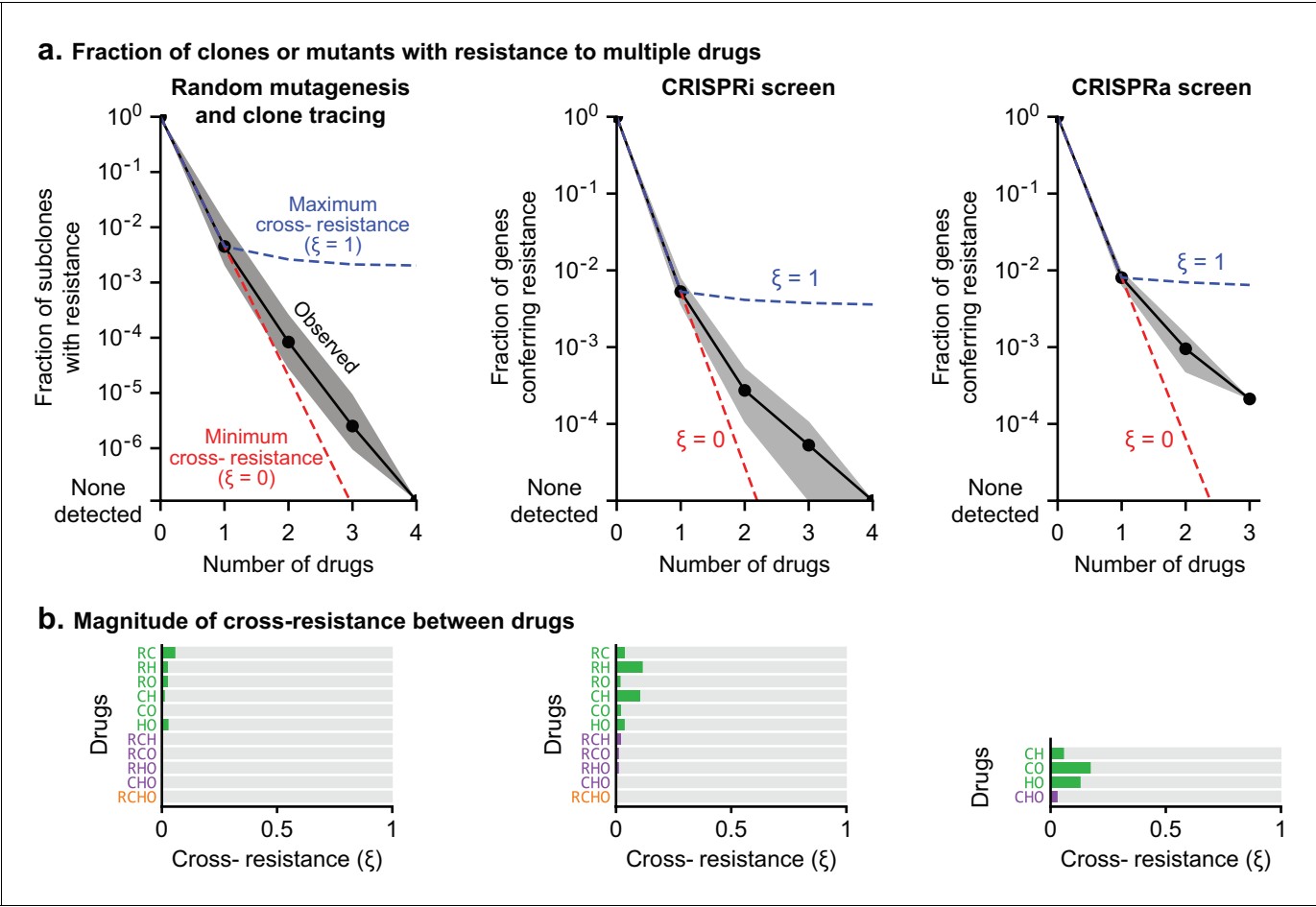

**Figure 8.** Cross-resistance between drugs in R-CHOP is close to its theoretical minimum. (**a**) Fraction of clones or genetic perturbations resistant to one or more drugs in RCHO. Gray shading spans the range for different sets of drugs (e.g. six different pairs), and black points mark the average on a log scale. Dashed blue line: average frequency of Multi-Drug Resistance (MDR) if resistance is maximally overlapping (maximal overlap is the minimum of constituent single drug MDR frequencies; cross resistance parameter ξ = 1). Dashed red line: average frequency of MDR as the product of single-drug MDR frequencies with ξ = 0. (**b**) Strength of cross-resistance (ξ) for different sets of drugs in RCHO, as determined from data summarized in (a).

should be discoverable using the methodology described here. We did identify multiple genes conferring hypersensitivity to two drugs, and three genes conferred hypersensitivity to three drugs (no genes were found that made cells hypersensitive to four drugs; *Figure 7—figure supplement 2B, D*). For example CRISPRa of *LMO2* sensitized cells to C and O; *LMO2* is highly expressed in the Germinal Center subtype of DLBCL, which responds better to R-CHOP than the *LMO2*-low Activated B-Cell subtype (*Alizadeh et al., 2000*). Across all subtypes, high *LMO2* expression is the strongest single gene predictor of survival in DLBCL patients treated with CHOP or R-CHOP (*Lossos et al., 2004*; *Natkunam et al., 2008*). Thus, the prognostic value of *LMO2* may be mediated in part through its impact on sensitivity to cyclophosphamide and vincristine. Hypersensitivity genes may be addressable therapeutically since 11% (23 from by CRISPRi and 49 from CRISPRa) are members of the 'liganded genome' – the subset of proteins for which high affinity small molecules are available (*Moret et al., 2019*; *Roberts et al., 2017*). Study of such compounds is beyond the scope of the current study but our data suggest possible avenues to enhancing responsiveness to R-CHOP in DLBCL.

## Discussion

The use of cancer drugs in combination was motivated historically by the need to overcome selection for drug resistance, which is a primary limitation on the durability of responses to monotherapy (*Law, 1952*). Inspired in part by multi-drug cures for tuberculosis, it was proposed that lasting remission required combined use of chemotherapies having different mechanisms of action and thus, different mechanisms of resistance (see retrospectives by *Frei and Antman, 2000*; *Schnipper, 1986*). Recent mathematical models of tumor evolution support these ideas, and predict that curing a cancer requires non-cross-resistant drug combinations in which the probability of acquiring resistance to all drugs is effectively zero (given the number of tumor cells and the magnitude of tumor cell killing) (*Bozic et al., 2013*). In contrast, contemporary drug development focuses on identifying synergistic pharmacological interactions among drugs when considering possible new combination therapies (*Al-Lazikani et al., 2012*; *Chou, 2010*; *Han et al., 2017*; *Lehár et al., 2009*; *Nature Medicine, 2017*; *Sun et al., 2013*).

In this paper we directly assess pharmacological interaction and cross-resistance among the drugs comprising R-CHOP, a highly successful curative cancer therapy developed over the course of decades by experimentation in patients with DLBCL. To investigate the pharmacological mechanisms underlying the success of R-CHOP we studied R-CHOP constituents individually and in combination in DLBCL cell lines. Over a wide range of drug concentrations, we observed no significant synergy among R-CHOP drugs using either Bliss Independence or Loewe Additivity criteria: pairwise drug interactions ranged from additive to antagonistic, and the combined activity of all five drugs was close to purely additive. The significance of antagonism among some drug pairs is not clear but studies of antibiotics have shown that antagonistic drug interactions can suppress the emergence of drug-resistant mutants (*Chait et al., 2007*; *Michel et al., 2008*; *Yeh et al., 2009*).

We investigated the frequencies and mechanisms of resistance to R-CHOP using DNA barcoding and CRISPRi/a technology. These three approaches made it possible to explore drug resistance and cross-resistance caused by point mutations, chromosome aberrations, and increased or decreased gene dosage. Large library size ($10^6$ clones) is a strength of the barcoding approach and it yielded $>10^4$ DLBCL mutants resistant to one or more drugs (as measured in biological triplicate). Genome-wide CRISPRi/a screening is more challenging technically, but it reveals the identities of genes involved in resistance and sensitivity. The results of all three resistance screens were clear: progressively fewer clones were observed with resistance to one, two, three or four drugs.

The specific frequencies of drug resistance reported in this study may not be directly relevant to the clinical setting, since such frequencies depend on experimental features such as population size, drug concentrations, and the inherent sensitivity of the cultures under study. Instead, the key finding is that the number of multi-drug resistant cells we observe is close to the theoretical minimum: with no cross-resistance, the number of cancer cells resistant to each of *n* drugs approximately equals $P_0 \prod_i^n f_i$, where $P_0$ is the initial cancer cell population and $f_i$ is the frequency of resistance to drug *i*. Our data therefore confirm the hitherto untested theories of Law and Frei (*Frei et al., 1965*; *Law, 1952*).

The existence of multi-drug resistance mechanisms (such as transporter overexpression) is responsible for the slight increase in cross-resistance over the theoretical minimum in cell line studies. However, the quantitative impact of transporters and similar genes on the overall frequency of multi-drug resistance is relatively modest because mutation or over-expression of such genes is substantially rarer than of genes conferring resistance to single drugs; moreover, the transporters we identified were active only on a subset of R-CHOP drugs, not all of them. In the clinical setting, a newly diagnosed patient often has $P_0 > 10^{10}$ lymphoma cells, and sensitivity to each drug in R-CHOP is expected to vary widely among patients due to innate resistance (speculatively, $10^{-6} < f_i \leq 1$, where complete resistance to drug $i$ corresponds to $f_i = 1$). Thus, even in the absence of acquired cross-resistance, R-CHOP does not cure all cases of DLBCL, a phenomenon we discuss below.

## Limitations of our analysis

The current work was performed in cultured cells and it is likely that the components of R-CHOP have additional mechanisms of action in human patients that we were unable to assay. For example, rituximab is cytotoxic to DLBCL by signaling-induced cell death, complement-mediated cytotoxicity (CMC), and antibody dependent cellular cytotoxicity (ADCC) (*Weiner, 2010*). In this study, we only scored CMC since DLBCL cultures displayed little direct induction of apoptosis, and ADCC reconstituted in vitro using peripheral blood mononuclear cells elicits insufficient cell death for selection of resistant clones (typically less than 50% killing) (*Dall'Ozzo et al., 2004*; *Reff et al., 1994*). We cannot exclude the possibility that ADCC interacts synergistically with chemotherapy but note that the immunosuppressive effects of many chemotherapies disfavor this hypothesis. With regard to the evolution of drug resistance, rituximab may behave as though it is several drugs in one due to its multiple mechanisms of action, although CD20 loss constitutes a shared cause of resistance (*Figure 9A*). Prednisone, the pro-drug of prednisolone, can induce remissions of DLBCL even as a single agent (*Lamar, 2016*), although cytotoxicity was not evident in cell cultures and therefore not adequately tested in our studies. It is also possible that synergistic or antagonistic interactions may occur by mechanisms specific to the in vivo setting; cell non-autonomous processes involving the immune system are one obvious possibility. R-CHOP is often successful in treating Acquired Immune Deficiency Syndrome (AIDS) related DLBCL, suggesting that adaptive immunity is not essential to its activity (*Ribera et al., 2008*). Cell culture is nonetheless an appropriate and necessary setting in which to test the Law and Frei hypothesis: such a test requires molecular manipulations and phenotypic screens over wide dose ranges that can only be performed in culture, and evidence that an animal model (or patient) benefits from combination therapy does not discriminate among alternative mechanisms of benefit. Moreover, the assessment of synergistic interactions in new drug combinations is most often performed in culture, making our analysis of interactions in R-CHOP relevant to current drug discovery efforts.

## The fractional kill hypothesis and patient-to-patient variability

In their 1964 study on the curability of experimental leukemia, Skipper, Schabel, and Wilcox observed that a given dose of alkylating chemotherapy kills a fixed fraction of cancer cells regardless of population size (equivalently, the logarithm of the number of cancer cells is reduced by a fixed quantity) (*Skipper et al., 1964*). 'Fractional kill', also called 'log kill', has been observed for many cancer therapies and is thought to reflect the impact of genetic and phenotypic heterogeneity on drug response (*Paek et al., 2016*; *Spencer et al., 2009*). Considered in the context of combination therapy, the number of 'log kills' contributed by each drug is expected to be arithmetically additive if the drugs have different resistance mechanisms, precisely as we have observed for R-CHOP (*Figure 9A*). For example, if each of two drugs elicit 99% kill (two log-kills), and the 1% surviving fractions overlap by no more than chance, as occurs with low cross-resistance, then only 1% of 1% of cancer cells will survive both drugs used in combination, resulting in 99.99% kill (2 + 2 = 4 log-kills). Thus, combinations of individually effective drugs with low cross-resistance can achieve high fractional tumor cell killing despite a lack of synergistic drug interaction.

Enhanced fractional kill can equivalently be understood from a pharmacological perspective. As discussed above, the dose response of DLBCL to R-CHOP is very close to additive, which is nonetheless a basis for therapeutic superiority over monotherapy. Because toxicities limit the maximum dose of each single agent, when the constituents of a combination have qualitatively different toxicities

**Figure 9.** The role of multiple drug mechanisms in increasing the probability of cure by combination therapy. (a) Conceptual schematic of the role of multiple drug mechanisms, each subject to different mechanisms of resistance, in the eradication of drug resistant clones and cure of a patient's cancer. When drug cross-resistance is low (among cancer cells in one patient), the 'log-kills' achieved by each drug mechanism add up. (b) Conceptual schematic of the role of patient-to-patient variability in drug sensitivity. The number of cancer cells that survive $n$ drugs is $P_0 \prod_i^n f_i$, where $P_0$ is initial population, and $f_i$ is fraction of cells that survive drug $i$ (on log-scale this is $\text{Log}(P_0) + \Sigma_i \text{Log}(f_i)$; note $\text{Log}(f_i) \leq 0$). The effect of combination therapy in each patient depends on initial tumor burden and the magnitude of effect of each constituent drug ($f_1$, $f_2$, etc), which varies across patients. Whether a patient is cured depends on the number of cancer cells surviving all drugs (vertical gray line), which can be zero in some patients and large in others, even if drugs are always additive and lack cross-resistance. Consistent with the clinical history of DLBCL, increasing the number of individually efficacious and non-cross-resistant drugs in combination therapies is expected to increase the fraction of patients cured; although an added drug could fail to improve efficacy if it lacks individual efficacy or is cross-resistant with drugs already given. It remains unresolved whether chemotherapy must eradicate every cancer cell.

(as in R-CHOP) (*Neal and Hoskin, 2009*) they can be administered simultaneously at close to maximum tolerated doses. The result of a higher 'sum of dose intensities' (*Frei et al., 1998*) is precisely what is predicted based on the fractional cell kill model, if it can be achieved at acceptable toxicity.

The results of this paper do not suggest that drug additivity and low cross-resistance are sufficient for cure: a critical feature of an additive drug combination is the absolute magnitudes of each drug's efficacy (or log-kills). Because of the large numbers of cancer cells often present in DLBCL at the time of diagnosis ($>10^{10}$), and because the goal of a therapy is to get below a critical number of surviving cancer cells – less than one for the sake of argument – it is logical to presume that each drug in R-CHOP must contribute 2–3 log kills, on average, to achieve a cure. It remains unknown whether a cure requires that a chemotherapy regimen achieve less than one viable cell or if the immune system can clear malignant diseases when cell number falls below some threshold above one (*Frei, 1972*). The conclusions in the current paper are agnostic with respect to this important issue.

How does patient-to-patient variability impact the success of therapies such as R-CHOP? In *Figure 9B*, we schematize the effect of both (i) inter-patient variability in drug response and (ii) addition of log-kills of different magnitudes. Across a population of patients, the absolute magnitude of each drug's effect is expected to vary (denoted in *Figure 9B* by lines of different lengths) as does initial tumor size (denoted by the differing origins of the lines). We have previously shown that inter-patient variability in sensitivity to different drugs explains some of the benefit of combination therapies, simply by increasing the probability of a good response to any one drug (*Palmer and Sorger,*

*2017*). In therapies such as R-CHOP, each patient's net response is the sum of single-drug responses, with each drug's contribution varying from one patient to the next. Any single drug could be ineffective in some patients, even among patients who might be cured by the combined effect of other drugs in their treatment. Increasing the number of individually effective drugs increases the probability that a patient will have multiple strong responses achieving a curative degree of fractional kill (denoted by the heavy green line). These concepts are consistent with the historical facts that some cases of DLBCL are curable with three drugs (CVP), more are cured with four drugs (CHOP) and yet more with five (R-CHOP); but even with the best available therapy some patients are still not cured. Improving survival further by adding new drugs to R-CHOP has proven challenging, and prior to the introduction of Ritixumab the 7-drug 'ProMACE-CytaBOM' regimen failed to improve DLBCL survival relative to CHOP (*Fisher et al., 1993*). The reasons for this are likely to be multi-faceted: some drugs in ProMACE-CytaBOM may be ineffective in many patients; they may have overlapping toxicities leading to dose reductions and interruptions that compromise efficacy (*Cabanillas, 2007*); and they may have overlapping resistance with standard therapies such that they fail to achieve greater fractional kill.

## Systematic study of drug cross-resistance

With respect to drug synergy in combination therapy, a fundamental limitation of current tests for pharmacological interaction is they pertain to doses near the $IC_{50}$ value and are therefore relevant to the most drug-susceptible part of a cell population. In contrast, the primary obstacle to cure in most settings is thought to be acquired drug resistance caused by rare resistance mutations, which can arise even at very high doses. Unfortunately, systematic analysis of cross-resistance is very difficult using conventional cell culture techniques (*Law, 1956*), perhaps explaining why the 'non-overlapping resistance' hypothesis has not been extensively explored in pre-clinical drug development. The introduction of clone tracing and genome-wide CRISPR technologies fundamentally changes the situation: using these methods, cross-resistance can easily be studied for any new combination therapy active in cultured cells. Moreover, since screening for resistance is best performed on individual drugs and cross-resistance identified in a subsequent computational comparison, information on resistance genes can be acquired cumulatively, making cross-resistance analysis scalable to many drugs and many combinations. Consistent with this idea, clone tracing has been proposed as a general approach to identifying combination regimens with non-overlapping resistance (*Bhang et al., 2015*). Our findings supports this proposal and demonstrate the additional advantages of CRISPRi/a. We propose that screening for cross-resistance should become as routine in pre-clinical cancer pharmacology as screening for pharmacological interaction.

## Materials and methods

**Key resources table**

| Reagent type (species) or resource | Designation | Source or reference | Identifiers | Additional information |
|---|---|---|---|---|
| Cell line (*Homo-sapiens*) | Pfeiffer | ATCC (Cat# CRL-2632) | RRID:CVCL_3326 | Diffuse Large B-Cell Lymphoma |
| Cell line (*Homo-sapiens*) | SU-DHL-4 | ATCC (Cat# CRL-2957) | RRID:CVCL_0539 | Diffuse Large B-Cell Lymphoma |
| Cell line (*Homo-sapiens*) | SU-DHL-6 | ATCC (Cat# CRL-2959) | RRID:CVCL_2206 | Diffuse Large B-Cell Lymphoma |
| Biological sample (*Homo-sapiens*) | Pooled human complement serum | Innovative Research | IPLA-CSER | |
| Peptide, recombinant protein | Rituximab | Dana Farber Cancer Institute | | 8 mg/mL in the clinical formulation + 10% Glycerol |
| Chemical compound, drug | 4-hydroperoxy-cyclophosphamide (4-H-Cyclo.) | Niomech | D-18864 | Pre-activated form of cyclophosphamide |

*Continued on next page*

*Continued*

| Reagent type (species) or resource | Designation | Source or reference | Identifiers | Additional information |
|---|---|---|---|---|
| Chemical compound, drug | Doxorubicin | Selleck Chemicals | S1208 | |
| Chemical compound, drug | Vincristine | Selleck Chemicals | S1241 | |
| Chemical compound, drug | Prednisolone | Selleck Chemicals | S1737 | Pre-activated form of prednisone |
| Commercial assay or kit | CellTiter-Glo | Promega | G7573 | Luminescent cell viability assay |
| Chemical compound, drug | N-ethyl-N-nitrosourea (ENU) | Sigma Aldrich | N3385 | Mutation-inducing agent |
| Recombinant DNA reagent | ClonTracer Barcoding library | Addgene (Cat# 67267) | RRID:Addgene_67267 | (*Bhang et al., 2015*) |
| Software, algorithm | clonTracer_analyze v1.0 | (*Bhang et al., 2015*) | | Script for analysis of barcode composition |
| Cell line (*Homo-sapiens*) | HEK293T | ATCC (Cat# CRL-3216) | RRID:CVCL_0063 | For lentivirus production |
| Recombinant DNA reagent | psPAX2 | Addgene (Cat# 12260) | RRID:Addgene_12260 | Lentiviral packaging plasmid |
| Recombinant DNA reagent | pCMV-VSV-G | Addgene (Cat# 8454) | RRID:Addgene_8454 | VSV-G envelope expressing plasmid for lentivirus production |
| Cell line (*Homo-sapiens*) | Pfeiffer CRISPRi | This paper | | See Materials and methods, Section 'Generation of dCas9-expressing cell lines' |
| Recombinant DNA reagent | pMH0001 | Addgene (Cat# 85969) | RRID:Addgene_85969 | Lentiviral construct for expression of dCas9-BFP-KRAB |
| Cell line (*Homo-sapiens*) | K562 | ATCC (Cat# CCL-243) | RRID:CVCL_0004 | Chronic myeloid leukemia (CML) cell line |
| Cell line (*Homo-sapiens*) | K562 CRISPRa | This paper | | See Materials and methods, Section 'Generation of dCas9-expressing cell lines' |
| Recombinant DNA reagent | pHRdSV40-dCas9-10xGCN4_v4-P2A-BFP | Addgene (Cat# 60903) | RRID:Addgene_60903 | Lentiviral construct for expression of dCas9-SunTag |
| Recombinant DNA reagent | pHRdSV40-scFv-GCN4-sfGFP-VP64-GB1-NLS | Addgene (Cat# 60904) | RRID:Addgene_60904 | Lentiviral construct for expression of scFv-sfGFP-VP64 |
| Recombinant DNA reagent | pU6-sgRNA EF1Alpha-puro-T2A-BFP | Addgene (Cat# 60955) | RRID:Addgene_60955 | Lentiviral construct for expression of sgRNAs |
| Recombinant DNA reagent | hCRISPRi_v2 | Addgene (Cat# 83969 and 83970) | RRID:Addgene_83969 | Genome-wide human library of sgRNAs for CRISPRi |
| Recombinant DNA reagent | hCRISPRa_v2 | Addgene (Cat# 83978 and 83979) | RRID:Addgene_83978 | Genome-wide human library of sgRNAs for CRISPRa |
| Software, algorithm | Screen Processing pipeline | (*Horlbeck et al., 2016*) | | https://github.com/mhorlbeck/ScreenProcessing |
| Cell line (*Homo-sapiens*) | Pfeiffer CRISPRa | This paper | | See Materials and methods, Section 'Validation of CRISPR screens' |

## Cell culture and chemotherapies

Diffuse large B-cell lymphoma (DLBCL) cell lines were obtained from the American Type Culture Collection (ATCC) and the Dana Farber Cancer Institute. The identity of Pfeiffer (ATCC CRL-2632) was validated by Promega GenePrint 10 small tandem repeat (STR) profiling. All DLBCL cell lines were grown in RPMI-1640 with 25 mM HEPES and 2 mM L-alanine-L-glutamine (GlutaMAX) (Gibco 72400), supplemented to 4.5 g/L D-glucose, 10% (v/v) heat inactivated fetal bovine serum (FBS) (Gibco 16140071), and penicillin/streptomycin (P/S) at final concentrations of 100 U/mL and 100 µg/mL, respectively (Corning 30–002 CI). For CRISPRi screens, Pfeiffer cells were grown in RPMI-1640 (Gibco 72400) supplemented with 15% (v/v) FBS and P/S. K562 cells were grown in RPMI-1640 (ATCC 30–2001) with 10 mM HEPES, 4.5 g/L D-glucose, 2 mM L-glutamine, 1 mM sodium pyruvate, and supplemented with 10% (v/v) FBS and P/S. HEK293T cells were grown in Dulbecco's modified Eagle medium (Corning 10–013) with 4.5 g/L D-glucose, 4 mM L-glutamine, 1 mM sodium pyruvate, and supplemented with 10% (v/v) FBS and P/S. All cell lines were grown at 37°C and 5% $CO_2$. Cells were tested for mycoplasma contamination using the MycoAlert mycoplasma detection kit (Lonza). When treating with rituximab alone or in combination, media was additionally supplemented with 5% (v/v) pooled human complement serum (HCS) (Innovative Research IPLA-CSER) to enable complement-mediated cytotoxicity. Cells were grown in vented tissue-culture treated polystyrene flasks. Cell density and viability was assessed during culture by a TC20 automated cell counter (Bio-Rad) with trypan blue; all cell densities reported here refer to the count of live cells with diameter between 8 and 24 µm. During culture before drug treatment experiments, DLBCL cells were maintained at the following densities: Pfeiffer between $3 \times 10^5$ and $15 \times 10^5$ cells/mL; SU-DHL-4 and SU-DHL-6 between $2 \times 10^5$ and $10^6$ cells/mL; with centrifugation and transfer to fresh media every 2 to 4 days.

Chemotherapies were obtained as follows: 4-hydroperoxy-cyclophosphamide (4-H-Cyclo.) from Niomech (D-18864), doxorubicin, vincristine, and prednisolone from Selleck (S1208, S1241, and S1737), and rituximab from Dana Farber Cancer Institute. Single-use aliquots of 4-H-Cyclo. were prepared in DMSO at −80°C, other chemotherapies were prepared in DMSO and stored at −20°C, and rituximab was prepared at 8 mg/mL in the clinical formulation plus 10% glycerol and stored at 4°C. DMSO was obtained from Sigma (D2650) and puromycin from Gibco.

## Measurement of drug-drug interactions

All drug interaction experiments were conducted in biological duplicates using two independent cultures of the same cell line. After being split from a common ancestor, cultures were propagated in parallel for at least one week before any experiment. Dose responses to single or multiple drugs were measured on DLBCL cells grown in sterile black polystyrene 384-well assay plates. Each well was inoculated with 30 µL of culture at density $10^5$ cells/mL, and promptly afterwards concentration gradients of drugs were added to wells by D300 digital dispenser (Hewlett-Packard). All chemotherapies were dispensed as DMSO solutions, while rituximab was prepared at 2.5 mg/mL with 0.05% (v/v) Triton X-100, with a 90 s incubation after pipetting into the print cassette for liquid to be drawn into the print head. At the highest dispensed concentration of rituximab, this conferred a final Triton X-100 concentration of 3 parts-per-million, which we confirmed did not by itself inhibit the growth of DLBCL cells. Wells on plate edges were filled but not used for any measurements. The drug dispensing arrangement of each plate was spatially randomized (and re-organized during data analysis); thereby any spatial bias across a plate becomes random error rather than systematic error across dose responses. Whole control plates of untreated cultures demonstrated no detectable row bias or column bias. Each plate contained >40 untreated wells in randomized locations (not on edges) that served as no-inhibition controls. Assay plates were incubated at 37 °C with 5% $CO_2$, inside containers humidified by sterile wet gauze. After 72 h, plates were removed from incubation and cooled at room temperature for 30 min, before automated dispensing (BioTek EL406) of 30 µL of CellTiter-Glo (1:1 dilution in phosphate buffered saline (PBS)) into each well. Following a 10 min incubation at room temperature, each well's luminescence was measured in a plate reader (BioTek Synergy H1). At the time of the 384-well plates' initial seeding, 1.5 mL cultures in 6-well plates were prepared from the same cell suspension, with separate cultures including or excluding 5% HCS. At the time of drug addition to plates, one of each such culture was harvested, and cell density was counted (Bio-Rad TC20 using trypan blue), and 72 hr later (at the time of CellTiter-Glo addition to

384-well plates) another such untreated 1.5mL culture was harvested and counted. From these density measurements we calculated the number of cell divisions occurring during the time of the assay, which was used during data analysis to determine Growth Rate (GR) metrics (*Hafner et al., 2016*). Specifically, we used GR = $\log_2$[(relative viability after treatment, according to CellTiter-Glo) × (cell number per μL of untreated control culture at t=72 h) / (cell number per μL of untreated control culture at t=0)] / $\log_2$[(cell number per μL of untreated control culture at t=72 h) / (cell number per μL of untreated control culture at t=0)] (*Figure 1—figure supplement 1D*). By this measure GR=1 indicates full, uninhibited growth, GR=0 indicates complete growth arrest, or that proliferation and death are in balance (final cell count = initial cell count), and GR<0 indicates net cytotoxicity (final cell count < initial cell count; note that we did not impose an asymptotic lower bound of -1 as described by Hafner et al (this would be computed as $2^{GR} - 1$). HCS slightly speeded the division rate of Pfeiffer in the absence of drugs (17% shorter doubling time), and slightly diminished Pfeiffer sensitivity to 4-H-Cyclo. Pairwise drug interactions (*Figure 1*) were measured over an 11×11 'checkerboard' of logarithmically-spaced drug concentrations (5 points per order of magnitude), with 5% HCS in media only in interactions with rituximab (for this reason 4-H-Cyclo. is less potent in its isobologram with rituximab). The concentration range for each drug was selected based on preliminary dose-ranging studies so as to span a range from no detectable effect on growth to 98% reduction in cell number relative to untreated control cells, which corresponds to growth arrest plus 90% cell killing. High-order drug interactions, including pairs (*Figure 2*), were measured over 14-point concentration gradients of one to five drugs, in all cases including 5% HCS so that drug sensitivity and drug-free cell division rate was consistent across conditions that would be compared in analysis. For these high-order interactions, each independent culture (biological replicate) was measured with cultures seeded into duplicate plates (plate-to-plate technical duplicates). Each of these four combinatorially complete drug response sets spanned two 384-well plates, which each contained a full set of single-drug gradients, and thus single-drug responses were in total measured in octuplicate. In the analysis, '100% luminescence' was defined on a per-plate basis by the interquartile mean of at least 50 drug-free wells within that plate (excluding edges). For isobologram analysis (*Figure 1*), the topology of drug response over the 11×11 checkerboards was smoothed by a nearest-neighbor median filter; this will apply no change to a monotonic response surface, and only smoothes data in cases of locally non-monotonic (that is, jagged) dose response. The absence of this filter changes no conclusions regarding interaction types but yields occasionally jagged isoboles. Fractional inhibitory concentrations (FICs) are calculated by comparing dose responses of drug combinations to dose responses of their constituent single drugs. Given a mixture of drugs at a dose that causes 50% killing, $FIC_{50}$ is the sum of each drug's concentration in that mix as a fraction of the single-agent doses producing the same effect: $FIC_{50} = \sum \frac{IC50_{drugincombination}}{IC50_{drugalone}}$. FIC=1 indicates Loewe additivity.

## Production of ClonTracer lentivirus

ClonTracer library was a gift from Frank Stegmeier (Addgene 67267). Lentiviral particles carrying ClonTracer were produced by calcium phosphate transfection of HEK293T cells (grown in DMEM with 10% FBS and 10 mM HEPES) with ClonTracer plasmid (10 μg per 10 cm dish) and lentiviral packaging and VSV-G plasmids psPAX2 and pMD2.G (Cellecta CPCP-K2A; 10 μg of mix per 10 cm dish). Supernatants of transfected HEK293T cells were harvested at 48 hr and again at 72 hr post-transfection. Supernatants were pooled and clarified by centrifugation (500 × g, 10 min). Lentiviral particles were concentrated from supernatant by mixing three parts supernatant with one part Lenti-X concentrator solution (ClonTech 631231), incubating overnight at 4°C, centrifuging at 1500 × g and 4°C for 45 min, removing supernatant, and resuspending pellet at 1/100 original volume in PBS.

## DNA barcoding of cell lines

$10^8$ Pfeiffer cells in complete media were treated with 100 μg/mL N-ethyl-N-nitrosourea (ENU) for 4 hr; this was previously determined to be the highest dose tolerable by Pfeiffer for this duration without conferring detectable cell death. Cells were washed twice and returned to drug-free media for 72 hr to recover. $10^7$ of these cells were infected with the ClonTracer lentiviral library by 'spinoculation'. In this protocol, five microcentrifuge tubes were prepared containing $2 \times 10^6$ cells in 1 mL complete media, with 8 μg/mL polybrene, and lentivirus at a volume yielding a multiplicity of infection (MOI) of 0.1 (per tube, this was 5 μL of 100 × concentrate of lentivirus containing supernatant;

see measurement of MOI below). Tubes were incubated for 10 min (37°C, 5% $CO_2$), and centrifuged at 800 × g and 37°C for 60 min. The supernatant was removed, and each cell pellet was resuspended in 4 mL complete media ($5 \times 10^5$ cells/mL) and transferred to one well of a 6-well plate for continued growth (incubation at 37°C, 5% $CO_2$). Volume of lentivirus to produce this MOI had been previously determined by test infections of Pfeiffer with different volumes of lentiviral solution, after which the fraction of infected Pfeiffer cells were counted by flow cytometric analysis of the red fluorescent protein encoded by the ClonTracer cassette (BD LSRII, ex:488 nm, em:575/26 nm), having first gated out dead cells (Violet Viability kit, Thermo Fisher Scientific L34958, ex:405 nm, em:450/50 nm) (note, red fluorescence was not readily detectable until 2 days post-infection). From the measured fraction of fluorescent cells, MOI was calculated assuming a Poisson distribution of infection events. Cells were expanded in complete media for 3 days before applying selection for infected cells (which carry a puromycin resistance gene in the ClonTracer cassette): 3 days in 0.25 μg/mL, 3 days in 0.5 μg/mL, and then 2 days in 1 μg/mL puromycin. At this time flow cytometry could not detect a significant population of non-fluorescent cells. Barcoded Pfeiffer cells were grown without puromycin for an additional 4 days before selection experiments in R-CHOP.

## Selection for drug-resistant clones

From a well-mixed suspension of barcoded Pfeiffer cells, $10^8$ cells were harvested and frozen for measurement of pre-treatment DNA barcode frequencies by sequencing. From the same suspension of cells and at the same time, 15 replicate cultures were prepared in 75 cm² flasks with 25 mL of complete media containing $5 \times 10^5$ cells/mL of barcoded Pfeiffer cells. The total count of $12.5 \times 10^6$ cells per flask is calculated to contain 99.999% of the $10^6$ unique clones assuming equal initial abundance and random assortment into flasks. Because rituximab displayed an 'inoculum effect' with limited cytotoxicity at high cell density, three cultures for rituximab treatment were prepared at lower density and higher volume: 60 mL of barcoded Pfeiffer at $1 \times 10^5$ cells/mL in 150 cm² flasks, in media supplemented with 5% HCS. The total count of $6 \times 10^6$ cells in each rituximab-treated flask is calculated to contain 99.7% of the $10^6$ clones. For each drug, and DMSO control, three replicate flasks were treated for 72 hr at the following concentrations: 4 μM 4-H-Cyclo.; 50 nM doxorubicin; 5.6 nM vincristine; 16 μg/mL rituximab; 0.04% (i.e., 0.0004 v/v) DMSO (the highest DMSO concentration delivered with any drug). These drug concentrations were chosen on the basis of preliminary dose-finding experiments that identified them to be the highest concentration, in a series of 2-fold concentration steps, from which any surviving cells repopulated the culture within 2 weeks of recovery following the 72 hr drug treatment. Following treatment, cultures were washed twice and resuspended in drug-free media. During recovery, culture volumes were adjusted to maintain cell density within the recommended range ($3–15 \times 10^5$ cells/mL). No cells were disposed of except from the DMSO control flasks, which suffered no inhibition but were maintained for a 'recovery' time to match drug-treated flasks. Following recovery to a population size twice the initial inoculum, the recovered cultures were exposed to repeat treatments (each flask treated by the same drug as before), and recovery. Following the second recovery, cultures were centrifuged and cell pellets harvested for barcode sequencing. Prednisolone treatments were designed differently because it was not cytotoxic to cultured Pfeiffer cells (nor any of six other DLBCL cell lines) in concentrations up to 50 μM. Therefore, triplicate Pfeiffer cultures (25 mL in 75 cm² flasks) were maintained in 20 μM prednisolone for 20 days, with cell density maintained between $3–15 \times 10^5$ cells/mL and with fresh prednisolone administered with media changes every 72 to 96 hr (*Figure 3—figure supplement 1A*). This treatment duration was estimated to produce ≈20-fold enrichment of clones fully resistant to the mild inhibitory effect of prednisolone (≈13 divisions in 20 days ⇒ enrichment from resisting 20% growth inhibition = $(1/0.8)^{13}$ = 18-fold).

## Barcode amplification and sequencing

To avoid contamination of pre-amplification materials with amplified DNA barcodes (which are approximately a billion-fold more concentrated), all materials, processes and equipment used prior to PCR amplification of ClonTracer barcodes were physically and temporally quarantined from all materials, processes and equipment used following PCR (distant benches and equipment, never both used on the same day). Genomic DNA (gDNA) was extracted from frozen cell pellets with DNeasy Blood and Tissue extraction kits (Qiagen 69504), using the spin-column protocol including

RNase A incubation. Four spin columns were used per sample of $10–15 \times 10^6$ cells; whereas the pre-treatment sample was a larger population of $3 \times 10^7$ cells applied to eight spin columns. DNA concentration was measured by SYBR green fluorescence with a λ dsDNA calibration curve (readings on BioTek Synergy H1). ClonTracer DNA barcodes consist of a repeating 'Strong (G or C) - Weak (A or T)' pattern with no detectable PCR amplification bias so that barcode counts measured by deep sequencing are proportional to clone abundance (*Bhang et al., 2015*). Barcodes were amplified from 20 µg of gDNA per sample, representing 3 million diploid genomes as template, with Q5 polymerase (New England Biolabs M0492). This was accomplished with parallel 50 µL reactions with 2 µg of template each. The pre-treatment sample was amplified from 40 µg of DNA, representing 6 million diploid genomes. Primer sequences were as described previously (see Supplemental Table 2 of *Bhang et al., 2015*). Reaction success and yield was verified by agarose gel electrophoresis. PCR products of all treatment conditions were pooled and size selected (133 bp) by excision from an agarose gel (using SYBR-safe stain and blue LED illumination) with purification by QIAquick Gel Extraction Kit (Qiagen 28704). PCR product from pre-treatment cell sample was processed separately rather than pooled with others. PCR products were sequenced on Illumina HiSeq 2500 in high-output single read mode, with custom read (CCGAGATACTGACTGCAGTCTGAGTCTGACAG) and index (AGCAGAGCTACGCACTCTATGCTAG) primers. A 30% PhiX spike-in provided necessary sequence diversity. FASTQ files were analyzed by the clonTracer_analyze v1.0 script which is available with the ClonTracer system (see www.addgene.org, cat. #67267). This script conducts the Barcode-composition analysis described by *Bhang et al. (2015)*, which identifies high-quality reads that conform to the expected barcode pattern (30nt of alternating weak (A/T) then strong (G/C)), and merges barcode sets that contain one high abundance barcode and sequence-adjacent barcodes (hamming distance 1 or 2) at much lower abundance indicating that they are sequencing errors of the high-abundance barcode. For each drug treated sample, $6–8 \times 10^6$ barcode reads were obtained, and from the pre-treatment sample $1.6 \times 10^8$ barcode reads were obtained; the latter being sequenced at greater depth.

## Analysis of barcode enrichment

Barcode counts in the pre-treatment sample were assigned a lower bound of the 5% quantile of counts in this sample (34 counts); this prevents barcodes that were rare or undetected in the pre-treatment sample from scoring as highly enriched in a drug treatment while having, for example, only two reads. Absolute barcode counts in pre- and post-treatment samples were then converted to the fraction of all counts for that sample. Each barcode's enrichment in a given drug treatment was calculated as post-treatment frequency divided by pre-treatment frequency. The biological triplicates of each treatment were merged to a single score by calculating the geometric mean enrichment. Each repeat was assigned a minimum enrichment of 1 when calculating geometric mean, to prevent severely penalizing barcodes that were not detected in one of three repeats; this is motivated by the statistical possibility that a barcode may be absent from any one flask's inoculum. A small fraction of barcodes exhibited geometric mean enrichment >1 in DMSO-treated cultures (1% of barcodes were enriched ≥10 fold), and therefore to normalize for these differences in fitness that are unrelated to drug sensitivity, we divided each barcode's enrichment scores in drug treatments by its enrichment in DMSO only when DMSO-enrichment was greater than 1 (enrichment scores in a drug treatment were not increased by having DMSO enrichment score less than 1).

## Lentivirus production for CRISPR reagents

HEK293T cells were transfected with the lentiviral plasmid of interest (as mentioned in relevant sections below), psPAX2 (Addgene #12260) and pCMV-VSV-G (Addgene #8454) in a 2:2:1 molar ratio using lipofectamine 3000 (Invitrogen) according to the manufacturer's instructions. The growth medium was replaced 6 hr post-transfection and was then harvested at 28 hr and 52 hr post-transfection. The two harvested growth medium fractions were pooled, centrifuged at $1,000 \times g$ for 10 min, and filtered through a 0.45 µm low-protein binding membrane. Lentivirus containing supernatants were stored at −80˚C. If needed, lentivirus titers were increased by adding ViralBoost reagent (Alstem) to the cell culture medium and lentivirus supernatants were concentrated using a lentivirus precipitation solution (Alstem).

## Generation of Cas9-expressing cell lines

To generate the Pfeiffer cell line stably expressing dCas9-KRAB (Pfeiffer CRISPRi), Pfeiffer cells (ATCC CRL-2632) were transduced with lentiviral particles produced using vector pMH0001 (Addgene #85969; expresses dCas9-BFP-KRAB from a spleen focus forming virus (SFFV) promoter with an upstream ubiquitous chromatin opening element) in the presence of 8 µg/mL polybrene. A pure polyclonal population of dCas9-KRAB expressing cells was generated by 3 rounds of fluorescence activated cell sorting (FACS) gated on the top half of BFP positive cells (BD FACS Aria II).

To generate the K562 cell line stably co-expressing dCas9 fused to the SunTag, and a SunTag-binding antibody fused to the VP64 transcriptional activator (K562 CRISPRa), K562 cells (ATCC CCL-243) were first transduced with lentiviral particles produced using vector pHRdSV40-dCas9-10xGCN4_v4-P2A-BFP (Addgene #60903; expresses dCas9 tagged with 10 copies of the GCN4 peptide v4 and BFP) in the presence of 8 µg/mL polybrene. After selection of BFP positive cells using one round of FACS (BD FACS Aria II), cells were transduced with lentiviral particles produced using vector pHRdSV40-scFv-GCN4-sfGFP-VP64-GB1-NLS (Addgene #60904; expresses a single chain variable fragment (scFv) that binds to the GCN4 peptide from the SunTag system, in fusion with superfolder green fluorescent protein (sfGFP) and VP64) in the presence of 8 µg/mL polybrene. Single cells with high GFP levels (top 25% of GFP positive cells) and high BFP levels (top 50% of BFP positive cells) were isolated by FACS and grown in single wells of a 96-well plate. Monoclonal cell lines were expanded and tested for their ability to increase the expression of target control genes (see section below). A single clone exhibiting robust growth and robust overexpression of target genes was selected as cell line K562 CRISPRa.

## Evaluation of CRISPRi/a cell lines using sgRNAs targeting individual genes

Pairs of complementary synthetic oligonucleotides (Integrated DNA Technologies) forming sgRNA protospacers flanked by BstXI and BlpI restriction sites were annealed and ligated into BstXI/BlpI double digested plasmid pU6-sgRNA EF1Alpha-puro-T2A-BFP (Addgene #60955). Oligonucleotides used to build sgRNA targeting individual genes are listed in *Supplementary file 1*. The sequence of all sgRNA expression vectors was confirmed by Sanger sequencing and lentiviral particles were produced using these vectors as described above (see 'lentivirus production'). Pfeiffer CRISPRi and K562 CRISPRa cells were infected with individual sgRNA expression vectors by addition of lentivirus supernatant to the culture medium in the presence of 8 µg/mL polybrene. Transduced cells were selected using puromycin (0.8 µg/mL for Pfeiffer and 2 µg/mL for K562) starting 48 hr post-transduction and over the course of 7 days with daily addition of the antibiotic. After 24 hr growth in puromycin-free medium, $1 \times 10^5$ cells were harvested and total RNA was extracted using the RNeasy Plus Mini kit (Qiagen). cDNA was synthesized from 0.1 µg total RNA using Superscript IV reverse transcriptase (Invitrogen) and oligo(dT)$_{20}$ primers (Invitrogen), following the manufacturer's instructions. Reactions were diluted 4-fold with H$_2$O and qPCR was performed in 10 µL reaction volume in 96-well plates using PowerUp SYBR Green PCR Master mix (Thermo Fisher Scientific), 2 µL diluted cDNA preparation, and 0.4 µM of primers. All qPCR primers are listed in *Supplementary file 1*. To calculate changes in expression level of target genes, all gene specific Ct values were first normalized to the Ct value of a reference gene (GAPDH) to find a ΔCt value. Log$_2$ fold changes in expression were then determined by the difference between the ΔCt value of targeting sgRNAs and that of a non-targeting negative control sgRNA (ΔΔCt).

## CRISPRi/a screens

Genome-wide libraries of sgRNAs from Addgene (hCRISPRi_v2: #83969 and #83970; hCRISPRa_v2: #83978 and #83979; a gift from Jonathan Weissman [*Horlbeck et al., 2016*]) were amplified in MegaX DH10B T1R cells (Invitrogen). These two libraries are provided as two sub-libraries each containing about 100,000 individual plasmids (five sgRNAs per gene). Sub-libraries (100 ng) were electroporated into MegaX DH10B T1R cells according to the manufacturer's instructions and the resulting transformed cells were plated on 10 × 150 mm LB/Ampicillin (100 µg/mL) Petri dishes. After 17 hr at 30℃, cells were scraped off the plates, washed with LB, and plasmid DNA was prepared from the cell pellet using the Plasmid Plus Maxi kit (Qiagen). Coverage for each sub-library was determined by serial dilution and colony counting, and was at least 5,000 × for each sub-library.

Lentiviral supernatant was prepared using an equimolar ratio of each sub-library plasmid for both the hCRISPRi_v2 and the hCRISPRa_v2 sgRNA libraries as described above ('lentivirus production') and was stored at −80˚C. The multiplicity of infection (MOI) of both preparations was determined by titration onto the target cell line and quantification of the percentage of BFP positive cells 2–3 days post-transduction by flow cytometry (BD Biosciences LSR II).

For CRISPRi screens, Pfeiffer CRISPRi cells ($2.5 \times 10^8$) were transduced with the hCRISPRi_v2 library lentivirus at an MOI of 0.4 in 250 mL culture medium + 8 µg/mL polybrene in $3 \times 225$ cm$^2$ cell culture flasks (Costar). 24 hr post-transduction, cells were harvested and resuspended in 400 mL fresh medium in $4 \times 225$ cm$^2$ cell culture flasks. Starting 48 hr post-transduction, the culture medium was exchanged daily and cells were maintained at $0.8 \times 10^6$/mL in puromycin (0.8 µg/mL) in 400–500 mL. After 5 days in puromycin, the proportion of BFP positive cells determined by flow cytometry increased from 37% to 90% of the fraction of viable cells. After recovery for 1 day in puromycin-free medium, the library cells were ready for initiation of parallel drug selections. First, a T0 sample of $6 \times 10^7$ cells was harvested and stored at −80˚C. Each screen was initiated using $6 \times 10^7$ cells at $0.4 \times 10^6$/mL in $2 \times 225$ cm$^2$ cell culture flasks. Vincristine (O), 4-hydroperoxy-cyclophosphamide (C), and doxorubicin (H) were added from $500 \times$ stocks in DMSO. A DMSO-only screen was used as an untreated control screen. Rituximab (R) was added from a 2 mg/mL stock in PBS and 5% (v/v) HCS was added to the growth medium. A screen with matching treatment of 5% (v/v) HCS was used as an untreated control screen for rituximab. For the duration of the screen, cells were maintained in $2 \times 225$ cm$^2$ cell culture flasks at a minimum concentration of $0.4 \times 10^6$/mL in 150 mL (minimum coverage of 300 cells per sgRNA) by exchanging the medium to fresh medium every 2 days. For drug treatment, cells were treated with pulses of drug for 3 days followed by exchange of the growth medium. O (5.0 nM final concentration) was added on day 0, day 7 and day 11; C (3.3 µM) was added on day 0 and day 3; H (27 nM) was added on day 0 and day 7; R (4 µg/mL and 5% HCS) was added on day 0, day 5 and day 10. During the course of the screen, cell count and viability were measured using a TC20 automated cell counter (Bio-Rad) using trypan blue. The vincristine CRISPRi screen underwent 7.60 fewer population doublings than the DMSO control screen; the 4-hydroperoxy-cyclophosphamide screen underwent 9.34 fewer doublings; and the doxorubicin screen underwent 7.41 fewer doublings. The rituximab CRISPRi screen underwent 7.53 fewer population doublings than the 5% HCS control screen. At day 14, $8 \times 10^7$ cells were harvested from each screen by centrifugation, washed twice with PBS and gDNA was extracted using the QIAamp DNA Blood Maxi Kit (Qiagen) according to the manufacturer's instruction, except that the elution was performed using 10 mM Tris·HCl pH 8.5. Typical yields from $8 \times 10^7$ cells ranged from 500 to 650 µg gDNA.

For CRISPRa screens, K562 CRISPRa cells ($3 \times 10^8$) were transduced with the hCRISPRa_v2 library lentivirus at an MOI of 0.25 in 300 mL culture medium + 8 µg/mL polybrene in $3 \times 225$ cm$^2$ cell culture flasks (Costar). 24 hr post-transduction, cells were harvested and resuspended in 450 mL fresh medium in $4 \times 225$ cm$^2$ cell culture flasks. Starting 48 hr post-transduction, the culture medium was exchanged daily and cells were maintained at $0.8 \times 10^6$/mL in puromycin (1.5–1.75 µg/mL) in 400–500 mL. After 5 days in puromycin, the proportion of BFP positive cells determined by flow cytometry increased from 26% to 96% of the fraction of viable cells. After recovery for 1 day in puromycin-free medium, the library cells were ready for initiation of parallel drug selections. First, a T0 sample of $8 \times 10^7$ cells was harvested and stored at −80˚C. Each screen was initiated using $6 \times 10^7$ cells at $0.4 \times 10^6$/mL in $2 \times 225$ cm$^2$ cell culture flasks. Vincristine (O), 4-hydroperoxy-cyclophosphamide (C), and doxorubicin (H) were added from $500 \times$ stocks in DMSO. A DMSO-only screen was used as an untreated control screen. For the duration of the screen, cells were maintained in $2 \times 225$ cm$^2$ cell culture flasks at a minimum concentration of $0.4 \times 10^6$/mL in 150 mL (minimum coverage of 300 cells per sgRNA) by exchanging the medium to fresh medium every 2 days. For drug treatment, cells were treated with pulses of drug for 3 days followed by exchange of the growth medium. O (35.0 nM final concentration) was added on day 0 and day 8; C (7.5 µM) was added on day 0 and day 8; H (27 nM) was added on day 0 and day 8. During the course of the screen, cell count and viability were measured using a TC20 automated cell counter using trypan blue. The vincristine CRISPRa screen underwent 8.80 fewer population doublings than the DMSO control screen; the 4-hydroperoxy-cyclophosphamide screen underwent 10.52 fewer doublings; and the doxorubicin screen underwent 9.87 fewer doublings. At day 15, $8 \times 10^7$ cells were harvested from each screen by centrifugation, washed twice with PBS and gDNA was extracted using the QIAamp DNA Blood Maxi

Kit according to the manufacturer's instruction, except that the elution was performed using 10 mM Tris·HCl pH 8.5. Typical yields from $8 \times 10^7$ cells ranged from 500 to 680 µg gDNA.

sgRNA barcode sequences were amplified by PCR using the extracted gDNA from either CRISPRi or CRISPRa screens as template and Phusion (NEB M0530) as polymerase. An equimolar mix of primers with stagger regions of different length (CC_LSP_025 to CC_LSP_032) was used as forward primer (to maintain sequence diversity in the common linker region for high-throughput sequencing purposes) and barcoded index primers (CC_LSP_033 to CC_LSP_040) were used as reverse primers. Reactions were composed of $1 \times$ HF buffer, 0.2 mM dNTPs, 0.4 µM forward primer mix, 0.4 µM indexed reverse primer, 0.5 µL Phusion, 1.5 mM MgCl$_2$, and 5 µg gDNA in a volume of 50 µL. After initial melting at 98°C for 30 s, the reactions were subjected to 24 cycles of heating at 98°C for 30 s, annealing at 62°C for 30 s and extension at 72°C for 30 s, and were followed by a final extension step at 72°C for 5 min. After verification of the PCR reaction success by agarose gel electrophoresis using SYBR safe stain (Thermo Fisher Scientific) on a single 50 µL reaction, 50% of the extracted gDNA for each screen (gDNA from $4 \times 10^7$ cells, corresponding to a coverage of 200×) was used as template in PCR reactions (typically 50–70 reactions per screen). After pooling all reactions from each single screen, the amplified sgRNA barcode PCR product (~240–250 bp) was purified by agarose gel electrophoresis using the QIAquick gel extraction kit (Qiagen). The concentration of individual libraries was quantified by fluorescence using the Qubit dsDNA high sensitivity assay kit (Thermo Fisher Scientific). Individual indexed libraries were mixed in equimolar ratio and were further purified using a QIAquick PCR purification kit (Qiagen). After determining accurate concentrations by quantitative PCR (qPCR) using the NEBnext library quant kit for Illumina (NEB), pooled libraries were sequenced on an Illumina HiSeq 2500 platform using a 50 bp single read on a high output standard v4 flow cell with a 15–20% PhiX spike-in. A total of 51–72 million reads were obtained for each indexed screen (minimum coverage of 250×).

The fastq sequencing files were analyzed using a Python-based ScreenProcessing pipeline previously reported by *Horlbeck et al. (2016)* (https://github.com/mhorlbeck/ScreenProcessing) with the following modification introduced due to the use of a mix of forward primers with a staggered region of different length. All reads were first processed using Cutadapt (*Martin, 2011*) to remove the linker sequence in front of the sgRNA barcode in each read (CTTGGAGAACCACCTTGTTG). To count the abundance of each sgRNA barcode in every sample, trimmed sequences were aligned to the library of protospacers present in the hCRISPRi_v2 or hCRISPRa_v2. Typically, 83–87% of the number of raw reads were aligned to the library of protospacers. The count files were next used to generate negative control genes, and calculate enrichment phenotypes and Mann-Whitney p-values as previously described (*Gilbert et al., 2014*; *Horlbeck et al., 2016*). To estimate technical noise in the screen, simulated negative control genes (the same number as that of real genes) were generated by randomly grouping 10 sgRNAs from the pool of ~4000 non-targeting control sgRNAs present in the libraries. The phenotypic effect of each sgRNA was quantified by the rho phenotype metric (*Kampmann et al., 2013*) which calculates the log2 fold change in abundance of an sgRNA between the treated and vehicle control samples, subtracting the equivalent median value for all 4000 non-targeting sgRNAs, and dividing by the number of population doubling differences between the treated and vehicle control populations. Similarly, untreated growth phenotypes ('gamma' phenotypes) can be calculated by a comparison of vehicle control and T0 samples; and 'tau' phenotypes can be calculated by a comparison of treated and T0 samples (*Gilbert et al., 2014*; *Kampmann et al., 2013*). For each gene (and simulated control gene), which is targeted by 10 sgRNAs, two metrics were calculated: (i) the mean of the strongest five rho phenotypes by absolute value, and (ii) the p-value of all 10 rho phenotypes compared to the 4000 non-targeting control sgRNAs (Mann-Whitney test). For genes with multiple independent transcription start sites (TSSs) targeted by the sgRNA libraries, the two metrics were calculated independently for each TSS and the TSS with the lowest Mann-Whitney p-value was chosen for further analysis. sgRNAs were required to have a minimum of 25 counts in at least one of the two conditions tested to be included in the analysis. To deal with the noise associated with potential low count numbers, a pseudocount of 10 was added to all counts. Genes that had less than eight sgRNA rho phenotypes were not included for further analysis. Read counts and phenotype scores for individual sgRNAs are available in the Supplemental Data. Gene-level phenotype scores and p-values are available in the Supplemental Data.

## CRISPRi cyclophosphamide hypersensitivity screen

The additional CRISPRi cyclophosphamide screen (*Figure 5—figure supplement 1B*) for identification of hypersensitive hits was performed and analyzed as described above with the following key modifications. Pfeiffer CRISPRi cells ($2 \times 10^8$) were transduced with the top five half library of hCRISPRi_v2 (Addgene #83969, that is five sgRNAs per gene) at an MOI of 0.3 in 200 mL culture medium + 8 µg/mL polybrene in $2 \times 225 \text{ cm}^2$ cell culture flasks. 24 hr post-transduction, cells were harvested and resuspended in 300 mL fresh medium in $3 \times 225 \text{ cm}^2$ cell culture flasks. Starting 48 hr post-transduction, the culture medium was exchanged daily and cells were maintained at $0.8 \times 10^6$/mL in puromycin (0.6 µg/mL) in 300–400 mL. After 3 days in puromycin, cells were recovered for 1 day in puromycin-free medium. A T0 sample of $5 \times 10^7$ cells was harvested and stored at −80°C. The CRISPRi screen was initiated using $2.5 \times 10^7$ cells at $0.4 \times 10^6$/mL in $1 \times 225 \text{ cm}^2$ cell culture flask. For the duration of the screen, cells were maintained in $1 \times 225 \text{ cm}^2$ cell culture flask at a minimum concentration of $0.4 \times 10^6$/mL in 62.5 mL (minimum coverage of 250 cells per sgRNA) by exchanging the medium to fresh medium every 2 days. 4-hydroperoxy-cyclophosphamide (2.5 µM final concentration) was added on day 0 and day 8. The cyclophosphamide CRISPRi screen underwent 4.38 fewer population doublings than the DMSO control screen. At day 15, gDNA was extracted from $5 \times 10^7$ cells from each screen and half of that was used as template in PCR reactions (coverage of 250×). A total of 31–34 million reads were obtained for each indexed sample and 81–82% of those reads were aligned to the reference library of protospacers. Negative control genes were generated by randomly grouping five sgRNAs from the pool of ~2000 non-targeting control sgRNAs present in the half-library. For each gene (and simulated control gene), which is targeted by five sgRNAs, two metrics were calculated: (i) the mean of the strongest three rho phenotypes by absolute value, and (ii) the p-value of all five rho phenotypes compared to the 2000 non-targeting control sgRNAs (Mann-Whitney test). Low count numbers were dealt with by adding a pseudocount of 1 to all zero counts. Gene ontology analysis was performed on the full output list of genes ranked by hypersensitivity score using GOrilla (*Eden et al., 2009*). The reported p-value is the enrichment p-value computed according to the GOrilla algorithm. The 'FDR q-value' represents the correction of the p-value for multiple testing hypothesis.

## Validation of CRISPR screens

Validation of the genome-wide screens was performed by building and characterizing individual knockdown (n = 9) and overexpression (n = 8) cell lines identified in the screens as single-drug or multi-drug resistant. The target sgRNA sequences were selected from the genome-wide sgRNA libraries based on the strength of the observed screen phenotype. sgRNAs were cloned into expression vectors, and cell lines were built and characterized as described above (see 'Evaluation of CRISPRi/a cell lines using sgRNAs targeting individual genes' and *Supplementary file 1* for oligonucleotide sequences). Knockdown sgRNA lentivirus particles were transduced into Pfeiffer CRISPRi cells and overexpression sgRNA lentivirus particles were transduced into K562 CRISPRa and into a polyclonal population of Pfeiffer CRISPRa. This population of cells was built in the same way as K562 CRISPRa (see 'Generation of Cas9-expressing cell lines') except that Pfeiffer cells (ATCC CRL-2632) were used and a polyclonal population of cells with high GFP (top 25%) and BFP (top 50%) levels was purified using one round of FACS.

The sensitivity of each cell line to R-CHOP drugs was determined by measuring a 13–17 point dose response curve for each drug in a similar way as described above (see 'Measurement of drug-drug interactions'). Data from two sets of biological replicates composed each of two technical replicates was analyzed to compute an IC50 and 95% confidence intervals, by fitting to a sigmoidal dose-response with the 'NonlinearModelFit' function in Wolfram Mathematica 12. Drug IC50s of unperturbed cells were determined from data of two independent non-targeting sgRNAs each measured in two biological replicates composed of two technical replicates (n = 8).

In order to detect differences in drug sensitivities of individual overexpression cell lines over a longer period of time than 3 days, co-culture competition experiments were performed. Individual overexpression Pfeiffer CRISPRa cell cultures were mixed at a 1:1 ratio with a Pfeiffer CRISPRa cell culture transduced with a non-targeting sgRNA control. Co-cultures were grown in 12-well plates in 1 mL complete medium over the course of 16 days, maintained at a density between $0.1–1.0 \times 10^6$/mL and were treated with pulses of R, C, H or O. Drug-treated co-cultures underwent significantly

less population doublings than a DMSO-treated control (between 4.5 and 6.5 less population doublings). Genomic DNA was isolated from samples of each co-culture taken at T = 0, and T = 16 days for each drug (or DMSO) selection (0.5–1.0 × $10^5$ cells) using the QiAamp DNA Mini kit (Qiagen 51304) (48 samples from eight co-cultures). The composition of each co-culture was determined by qPCR amplification of the sgRNA barcodes in each gDNA sample using sgRNA-specific forward primers and a common reverse primer (see *Supplementary file 1*). All nine pairs of primers were specific to their target sgRNA barcode as no unspecific amplification of other barcodes was observed and qPCR amplification was linear (efficiency: 0.9–1.1). Each co-culture gDNA sample was probed with a primer pair against the targeting sgRNA barcode and a primer pair against the non-targeting sgRNA barcode. The ratio of the two cell lines was determined from the difference in $C_T$ values. Any change in fitness due to individual gene overexpression is detected by comparison of the DMSO sample at T = 16 days and the sample at T = 0. Changes in drug resistance due to individual gene overexpression are detected by comparison of drug-treated samples at T = 16 days and DMSO samples at T = 16 days. Values reported ($N_{mutant}/N_{control}$) are from three technical replicates of the qPCR quantification and are determined by the difference in $C_T$ values between the targeting and non-targeting sgRNA between drug-treated and DMSO-treated samples.

## Cross-resistance analysis of CRISPR screens

For each gene, a single aggregate resistance score was calculated by multiplying the two metrics determined in the screen processing pipeline (resistance score = $-\log_{10}$(Mann-Whitney p-value)× mean of the strongest five rho phenotypes). Genes required eight or more observed sgRNA rho phenotype scores in a specific screen for inclusion in the analysis. In order to account more accurately for the technical noise in the screen, 10 random sets of ≈19,000 simulated control genes were generated (matching the number of actual gene targets). A resistance score was then calculated for each simulated control gene in all 10 sets for all the drugs tested (four for CRISPRi and three for CRISPRa). Genes that have a resistance score above a specific cutoff in at least two conditions tested are defined as 'cross-resistant'. The cutoff for cross-resistance analysis was determined by systematically quantifying the number of simulated control genes that would score as cross-resistant over the full range of resistance score cutoffs (in 0.01 increments). We selected a cutoff that scored on average over the 10 sets of control genes a single (or less than one) double resistant negative control simulated gene over all possible two drug combinations. Cross-hypersensitivity analyses were performed in an analogous way (the hypersensitivity score was calculated in the same way as the resistance score).

## Quantifying cross-resistance in clone tracing and CRISPR screens

Strength of cross-resistance for different sets of drugs was quantified as a weighted sum of the maximum and minimum cross-resistance scenarios. Given rates of single-drug resistance $10^{-A}$ and $10^{-B}$ to drugs *a* and *b*, the theoretical minimum rate of cross-resistance is $10^{-A-B}$ (this scenario is indicated by cross-resistance parameter ξ = 0), and the theoretical maximum rate of cross-resistance is min($10^{-A}$ or $10^{-B}$) (this scenario is indicated by cross-resistance parameter ξ = 1). Given an observed frequency of multi-drug resistance (MDR), cross-resistance ξ is computed as the solution to the equation: MDR frequency = ξ × minimum($10^{-A}$ or $10^{-B}$) + (1 − ξ)× $10^{-A-B}$. This solution is ξ = ($10^{-A-B}$ − MDR) / ($10^{-A-B}$ − minimum($10^{-A}$ or $10^{-B}$)).

## Acknowledgements

We thank F Stegmeier and C Bhang for ClonTracer, A Letai and A Eberly Puleo for rituximab, and also M Chung, L Albacker, L Maliszewski, M Cokol, S Chopra, and K Subramanian for helpful discussions.

## Additional information

### Competing interests

Peter K Sorger: is a member of the SAB or Board of Directors of Merrimack Pharmaceuticals, Glencoe Software, Applied Biomath and RareCyte Inc and has equity in these companies. In the last five

years the Sorger lab has received research funding from Novartis and Merck. PKS declares that none of these relationships are directly or indirectly related to the content of this manuscript. The other authors declare that no competing interests exist.

## Funding

| Funder | Grant reference number | Author |
| --- | --- | --- |
| National Institutes of Health | P50-GM107618 | Adam C Palmer<br>Christopher Chidley<br>Peter K Sorger |
| National Institutes of Health | U54-CA225088 | Adam C Palmer<br>Christopher Chidley<br>Peter K Sorger |
| National Health and Medical Research Council | 1072965 | Adam C Palmer |
| James S. McDonnell Foundation | 2012036 | Adam C Palmer |

The funders had no role in study design, data collection and interpretation, or the decision to submit the work for publication.

## Author contributions
Adam C Palmer, Christopher Chidley, Conceptualization, Formal analysis, Investigation, Methodology, Writing—original draft, Writing—review and editing; Peter K Sorger, Conceptualization, Supervision, Funding acquisition, Writing—original draft, Writing—review and editing

## Author ORCIDs
Adam C Palmer ⓘ https://orcid.org/0000-0001-5028-7028
Christopher Chidley ⓘ http://orcid.org/0000-0002-8212-3148
Peter K Sorger ⓘ https://orcid.org/0000-0002-3364-1838

## Decision letter and Author response
Decision letter https://doi.org/10.7554/eLife.50036.sa1
Author response https://doi.org/10.7554/eLife.50036.sa2

# Additional files

## Supplementary files
• Supplementary file 1. List of oligonucleotides used in this study and sequence of sgRNA protospacers used in individual CRISPRi/a cell line construction.

• Transparent reporting form

## Data availability
All data generated during this study are included in the manuscript and supporting files. Source data is provided for all clone tracing and CRISPR screen experiments.

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
