## [Decision Letter]

**Acceptance summary:**

The work addresses an important question in the field of combination cancer therapy: whether successful treatment requires that the individual drugs in combination have synergistic activity, as is often assumed. Using three experimental approaches (drug interaction analysis, clone tracing and CRISPRa and CRISPRi screening), your group provides compelling evidence that drug synergy is not required. Rather, low levels of cross-resistance and additivity between the drugs are sufficient for cure, as demonstrated using Diffuse Large B-Cell Lymphoma and R-CHOP combination therapy as a model. As highlighted in the discussion, it will be interesting to extend this work to the in vivo setting where multiple immuno-oncology combinations are now being investigated.

**Decision letter after peer review:**

Thank you for submitting your article "A curative combination therapy achieves high fractional cell killing through low cross-resistance and drug additivity" for consideration by *eLife*. Your article has been reviewed by two peer reviewers, and the evaluation has been overseen by Charles Sawyers as the Reviewing Editor and Eduardo Franco as the Senior Editor. The reviewers have opted to remain anonymous.

The reviewers have discussed the reviews with one another and the Reviewing Editor has drafted this decision to help you prepare a revised submission.

Summary:

Palmer and colleagues use three experimental approaches: drug interaction analysis, clone tracing and CRISPR screening, to argue that a successful combination therapy does not require drug synergy, as is commonly thought, but low levels of cross-resistance between the drugs. They demonstrate this principle using R-CHOP, a combination therapy that is often successful in curing Diffuse Large B-Cell Lymphoma. All three reviewers found the topic to be thought-provoking and the experimental approach to be clever with convincing results. We will consider the manuscript for publication if you can address the points below in a revision.

Essential revisions:

1) More precise definition of synergy versus additivity:

As the notion of drug synergy (as well as additivity and antagonism) is at the very core of the paper, a discussion of the precise definitions of synergy, additivity and antagonism in the Introduction or at the beginning of Results would be very useful. Currently the authors are mostly saying they are using Bliss independence and Loewe additivity criteria, with very loose definitions of what they mean by these two. As many readers may not be very familiar with these, precise mathematical descriptions of these two criteria and of all the steps in producing plots in Figure 1 would significantly increase the understanding of the paper.

Another dimension according to which drug(s) are measured is efficacy: a discussion of how their findings relate to individual and drug combination efficacy would be very helpful. Also typical concentrations of the drugs (at which they are effective, but not too toxic) should be shown on the plots in Figure 1.

2) Resistance experiments were not done using the entire combination:

The authors looked for resistance mutations to a drug combination by applying drugs individually and identifying DNA barcodes of mutant cells that were significantly enriched in two or more conditions. A very small number of cells in a single resistant clone may be lost purely due to stochastic drift, so not all resistant mutants may be picked up this way. The authors conclusions would be strengthened if they also applied the entire drug combination (R-CHOP) and determined whether any (and which) mutant cells were significantly enriched in that setting, but it may be challenging to address the effects of prednisone and Rituxan in vitro. We do not expect you to rerun the experiment with R-CHOP but ask that you should address the lack of this data in the text.

3) Implications for in vivo setting, where synergy may be relevant:

The authors sufficiently describe issues related to the use of rituximab and prednisone in cell culture studies. However, they should also note that mechanisms of synergy may also be specific to the in vivo setting. For example, immune stimulation by 4-H-Cyclo or doxorubicin may promote Rituximab or prednisone killing.

4) Revisions to the model figure:

The "model" figure is somewhat problematic given the data in this manuscript and the previous Cell manuscript by these authors. Indeed, two models seem to fir the clinical data. A) Combined fractional kill by each drug results in tumor eradication (each drug contributes to overall tumor cell death) or B) Only one or two drugs is relevant for any given patient, and the more drugs that are used, the greater chance of including drugs relevant for a given patient (the argument posed in the previous Cell paper). The authors should attempt to reconcile these arguments.

Additionally, it is unclear what the authors mean by "cross-resistance" in the model figure. Are the authors referring to variations between patients in relative drug response or heterogeneity within a given patient? While the resistance studies in this work provide interesting data regarding the potential landscape of tumor relapse following therapy, but relatively little is shown regarding the development of resistance in the context of multi-drug regimens.

---

## [Author Response]

Essential revisions:1) More precise definition of synergy versus additivity:As the notion of drug synergy (as well as additivity and antagonism) is at the very core of the paper, a discussion of the precise definitions of synergy, additivity and antagonism in the Introduction or at the beginning of Results would be very useful. Currently the authors are mostly saying they are using Bliss independence and Loewe additivity criteria, with very loose definitions of what they mean by these two. As many readers may not be very familiar with these, precise mathematical descriptions of these two criteria and of all the steps in producing plots in Figure 1 would significantly increase the understanding of the paper.

This is an excellent point. The Results section now includes the precise – and straightforward – mathematical definitions of Bliss independence and Loewe additivity (Results section paragraph three). Noting that Loewe’s original definition was solely graphical (illustrated in Figure 1B inset), the revised results also presents the equation used to quantify Loewe-type synergy/antagonism as a Fractional Inhibitory Concentration also known as *Combination Index*.

Another dimension according to which drug(s) are measured is efficacy: a discussion of how their findings relate to individual and drug combination efficacy would be very helpful. Also typical concentrations of the drugs (at which they are effective, but not too toxic) should be shown on the plots in Figure 1.

The Results section has been revised to note that efficacy, defined here as the magnitude of cell kill at a given drug doses, is assessed by the Bliss independence model of drug interactions. The discussion draws attention to our conclusion that *efficacy* is of critical importance in evaluating drug combinations.

Figure 1 now denotes, along drug axes, the locations of “Cmax” doses, or peak drug concentrations measured in human serum following clinically relevant protocols (this data had previously been shown only in Figure 2—figure supplement 1). Cmax values lie at the upper end of the experimentally tested ranges, which we consider appropriate given that Cmax is by definition the peak in concentration. Rituximab is the only case in which Cmax is much greater than the experimental range at which we observed potent killing.

2) Resistance experiments were not done using the entire combination:The authors looked for resistance mutations to a drug combination by applying drugs individually and identifying DNA barcodes of mutant cells that were significantly enriched in two or more conditions. A very small number of cells in a single resistant clone may be lost purely due to stochastic drift, so not all resistant mutants may be picked up this way. The authors conclusions would be strengthened if they also applied the entire drug combination (R-CHOP) and determined whether any (and which) mutant cells were significantly enriched in that setting, but it may be challenging to address the effects of prednisone and Rituxan in vitro. We do not expect you to rerun the experiment with R-CHOP but ask that you should address the lack of this data in the text.

We have expanded a previously brief discussion of this important matter in the Results section (illustrated in Figure 3B), noting limitations that arise from practical experimental considerations, and their impact on interpretation.

We note that extensive higher-order drug interaction studies (as measured by cell killing) were performed with 2-5 drug combinations assayed at fixed dose ratios (Figure 2) so the reviewer’s concern applies only to CRISPR and barcoding studies.

To summarize, the single-drug barcoding experiments were conducted at the highest drug concentrations that left surviving cells capable of regrowth. This was necessary for the obvious reason that we require regrowth to allow cells to be recovered for barcode sequencing. Two-fold increases in drug concentration were observed to leave no survivors, which we think is related to the observation that DLBCL cultures very rarely survive attempts at single-cell cloning (in the absence of chemotherapy). The reasons for this are therefore likely to be unrelated to drug-mediated cell killing and instead, likely reflect autocrine conditioning of media or some other requirement for a minimum number of cells in a well.

Thus, a ‘full-dose’ combination that achieves a greater fractional kill than monotherapy, and reduces the number of surviving cells to the point at which ‘single cell cloning’ is a problem, would be scored as an absence of multi-drug resistance. In this case, H_0_ would be rejected for what are likely to be artefactual reasons. An alternative approach could be to apply the combination at lower dosage, to impose an overall similar fractional kill. Figure 3B is intended to illustrate why in a lower-dosage combination that leaves measurable survivors, barcode enrichment could equally result from modest multi-drug resistance, or strong resistance only to one drug in the combination. Thus a measurement of barcode enrichment after a treatment with multiple drugs cannot clearly distinguish between multi-drug resistant clones and single-drug resistant clones. For these reasons we believe that the most technically robust way to measure cross-resistance between drugs is to profile resistance to every drug individually. This, and not any technical complexity, is the reason we did not test the full R-CHOP combination at high dose.

We have edited the results to note these technical points and to point out that stochastic drift in single-drug experiments may cause an underestimate of both single- and multi-drug resistance frequencies. The revision also notes aspects of experimental design, including our use of large populations and biological triplicate experiments, that are intended to minimize the loss of meaningful signals due to stochastic drift.

3) Implications for in vivo setting, where synergy may be relevant:The authors sufficiently describe issues related to the use of rituximab and prednisone in cell culture studies. However, they should also note that mechanisms of synergy may also be specific to the in vivo setting. For example, immune stimulation by 4-H-Cyclo or doxorubicin may promote Rituximab or prednisone killing.

This is an excellent point that we addressed inadequately. The reviewers are correct that drugs working via immune mechanisms cannot be studied in cell monoculture. The discussion has been revised to note that there may be mechanisms of synergy or antagonism specific to the in vivo setting, particularly those that may derive from immunological effects that we cannot measure in cell culture.

4) Revisions to the model figure:The "model" figure is somewhat problematic given the data in this manuscript and the previous Cell manuscript by these authors. Indeed, two models seem to fir the clinical data. A) Combined fractional kill by each drug results in tumor eradication (each drug contributes to overall tumor cell death) or B) Only one or two drugs is relevant for any given patient, and the more drugs that are used, the greater chance of including drugs relevant for a given patient (the argument posed in the previous Cell paper). The authors should attempt to reconcile these arguments.

We thank the reviewers for pointing out this issue. The impact of patient-to-patient variability and independent drug action explored in our Cell paper and the additive kill model in the current paper *are* conceptually compatible, and while they do not always manifest in the same clinical scenarios, the revised manuscript presents a reconciliation of these concepts (Figure 9). We summarize below this reconciliation as explained in the revised text.

In our prior study we found that patient-to-patient variability in ‘best single agent response’ (reviewer’s point B) was able to explain increased response rates and prolonged PFS achieved with combinations used to treat essentially incurable metastatic carcinomas. However, ‘best single agent response’ is insufficient to explain cures, because single chemotherapies cure almost no cases of advanced cancer. In curative therapies, multiple agents must be active in each patient and they must achieve an effect superior to the best monotherapy; the data in the current paper argues that for R-CHOP the nature of this superiority is that the drugs are additive with respect to cell killing (not synergistic).

In summarizing this concept, Figure 9A illustrates drug additivity, free of the complications of patient variability. It represents the simplest way in which active agents can ‘add up’ on a ‘log-kill’ scale.

The revised Figure 9B introduces the complication of patient-to-patient variability in sensitivity to each drug. In this case, multiple drugs are active in each patient but the magnitude of activity is variable across patients. In some cases, we hypothesize, patients may even receive no benefit from some of the components of the combination. In principle though, the success of a multidrug regimen does not require that every drug be active in all patients; instead, it need only be true that patients experience sufficient activity from the set of drugs that will ‘add up’ to a curative fractional kill. Why some drugs act in an independent manner and others are additive is not addressed by the current experimental study of R-CHOP. We speculated at the end of our Cell article that drugs may be additive or synergistic in the subset of patients in which both are significantly active, but in the metastatic cancers analyzed in that study, response rates were generally low and too few patients were responsive to *both* drugs for additivity to be statistically detectable in a trial. Whether this is a correct or sufficient explanation is unknown.

We are working to formally codify these concepts in a future (theory) manuscript but hope the explanation above suffices, in combination with revisions to Figure 9B and changes made to the text.

We appreciate the reviewers’ exhortation to better explain the reconciliation of these concepts and hope to have achieved this with the revision.

Additionally, it is unclear what the authors mean by "cross-resistance" in the model figure. Are the authors referring to variations between patients in relative drug response or heterogeneity within a given patient? While the resistance studies in this work provide interesting data regarding the potential landscape of tumor relapse following therapy, but relatively little is shown regarding the development of resistance in the context of multi-drug regimens.

We apologize for this confusion. The caption of Figure 9 has been revised to specify that in this context we refer to heterogeneity within a given patient; low cross-resistance in this context results in addition of ‘log-kills’.